# λ-SecAgg: Partial Vector Freezing for Lightweight Secure Aggregation in Federated Learning

## Abstract

Secure aggregation of user update vectors (e.g. gradients) has become a critical issue in the field of federated learning. Many Secure Aggregation Protocols (SAPs) face exorbitant computation costs, severely constraining their applicability. Given the observation that a considerable portion of SAP's computation burden stems from processing each entry in the private vectors, we propose **P**artial **V**ector **F**reezing (**PVF**), a portable module for compressing computation costs without introducing additional communication overhead. **λ-SecAgg**, which integrates SAP with PVF, "freezes" a substantial portion of the private vector through specific transformations, requiring only $\frac{1}{\lambda}$ of the original vector to participate in SAP. Eventually, users can "thaw" the public sum of the "frozen entries" by the result of SAP. To avoid potential privacy leakage, we devise Disrupting Variables Element for PVF. We demonstrate that PVF can seamlessly integrate with various SAPs and it poses no threat to user privacy in the semi-honest and active adversary settings. We include 7 baselines, encompassing 5 distinct types of masking schemes, and explore the acceleration effects of PVF on these SAPs. Empirical investigations indicate that when $\lambda = 100$, PVF yields up to $99.5\times$ speedup and up to $32.3\times$ communication reduction.

## 1 Introduction

Machine learning technologies are applied in countless fields to improve service performance. However, aggregating large amounts of data for big data mining raises concerns regarding data privacy (Liu et al., 2021). *Federated Learning* (FL) (McMahan et al., 2017) keeps original data on the local devices while only requiring data owners to submit local training updates to a central server. Nonetheless, as Zhu et al. (2019) and Geiping et al. (2020) indicate, attackers can infer a user's local data by reversing the submitted updates. To address this issue, numerous research efforts have been focusing on *Secure Aggregation Protocols* (SAPs) (Liu et al., 2022b) for aggregating all user's model information while preserving individual privacy.

The widely discussed SAPs are based on *Secure Multi-Party Computation* (SMPC) (Xu et al., 2022; Sotthiwat et al., 2021), *Mask* (Bonawitz et al., 2017), *Homomorphic Encryption* (HE) (Aono et al., 2017) and *Differential Privacy* (DP) (Wei et al., 2020). For most SAPs, except for DP-based ones, the computation overhead often scales proportionally with the length of the model update vectors since most of these schemes involve masking (encrypting) each entry of the vector sequentially. Therefore the **computation** time for both masking and unmasking always experiences a steep escalation with the increase in vector length, as PracAgg (Bonawitz et al., 2017) in Figure 1, significantly constraining real-world applications. Especially in recent applications that utilize FL to fine tune *Large Language Models* (LLMs) (Ye et al., 2024) with billions of parameters, the computational overhead brought by SAP is unbearable.

On the other hand, DP-based solutions although have the best efficiencies, many studies (Stevens et al., 2022; Wang et al., 2021) posit that the minimal noise added by DP is insufficient to thwart attacks such as gradient inversion aimed at stealing users' local data, where adversaries can recover flawed but recognizable handwritten digit image (Wang et al., 2021). Therefore the security of DP in

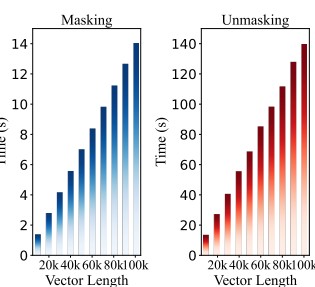

Figure 1: Computation time for one round, with 100 users and 10% dropout rate.

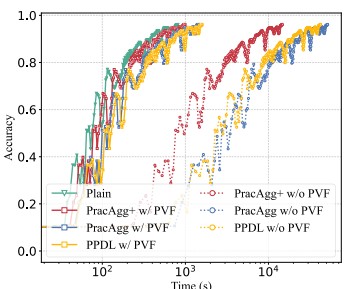

Figure 2: Comparison of various aggregation methods, with $n = 100, \eta = 5\%, \lambda = 100$.

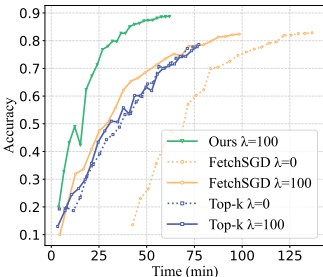

Figure 3: Comparison with compression-based techniques, with $n = 100, \eta = 5\%$.

secure aggregation faces challenges, necessitating its combination with masking to bolster privacy. In this work, we primarily explore methods to alleviate **masking-related** overhead.

**Our proposal.** We think the root cause of computation overhead in SAPs is masking each entry of the original vector. While a few *sparsification*-related approaches (Ergun et al., 2021; Lu et al., 2023) try to reduce the dimensions of uploaded vectors, they raise an inevitable trade-off of discarding some information. In this work, we propose **P**artial **V**ector **F**reezing (**PVF**) to reduce the number of entries processed in SAPs while ensuring intact aggregation of all entries in the original vector. Within the module, each user performs certain transformations on the original vector at a negligible computation cost to selectively *freeze* most entries of the user's original vector, compressing the length of the vector involved in SAPs to $\frac{1}{\lambda}$ of its original size ($\lambda$ is the compression factor). The communication overhead and number of communication interactions in SAPs after integrating PVF, which we call **$\lambda$-SecAgg**, do not increase. Further, we propose *Disrupting Variables Element* to prevent PVF from leaking the linear relationship between vectors. PVF remains decoupled from SAPs and guarantees individual user privacy under semi-honest and active adversary settings, offering high portability.

Our contributions can be summarized as follows:

- We propose the PVF without incurring additional communication overhead or harming security. It reduces the entries processed in SAP while **ensuring the aggregation of all entries in the original vector**, which means it can compress the computation overhead of SAP to approximately $\frac{1}{\lambda}$ of the original. Moreover, it brings up to $32.3 \times (\lambda = 100)$ additional communication enhancements for HE-based SAPs thanks to the decreased number of ciphertext entries.

- We propose the disrupting variables element to PVF to avoid potential privacy leakage.

- Extensive experiments show the effectiveness of our proposal. We include 7 baselines encompassing 5 types of **masking schemes** for a comprehensive overhead comparison, which is largely unexplored in most research endeavors and reaffirms the high portability of PVF.

## 2 RELATED WORK

**Secure Aggregation Protocols.** Various types of SAPs have been proposed, including SMPC-based (Boer & Kramer, 2020; Xu et al., 2022), HE-based (Aono et al., 2017; Ma et al., 2022; Li et al., 2022), DP-based (Geyer et al., 2017; Wei et al., 2020), and Mask-based (Bonawitz et al., 2017) schemes. Most efforts to reduce computation cost focus on enhancing PracAgg (Bonawitz et al., 2017), which is mainly categorized into two types: (i) improving the masking mechanism (Liu et al., 2022a; Stevens et al., 2022; Liu et al., 2022c; Wei et al., 2023) to reduce **masking-related** overhead; (ii) minimizing **interaction-related** overhead, including refining communication structures (Bell et al., 2020; So et al., 2021) and enhancing efficiency in key agreements among users (Kalikinkar et al., 2018; Kadhe et al., 2020; Ma et al., 2023). **Note** that the security of FL remains an open issue. SAPs, though cannot fully guarantee FL security at the moment, remain a promising direction worth exploration. The main objective of our work is to reduce the masking-related overhead of secure aggregation, thereby making it more applicable in practice.

**Compression-based Techniques**. Rothchild et al. (2020) employs a *Count Sketch* to compress model updates. Additionally, some sparsification-based approaches (Ergun et al., 2021; Lu et al., 2023) can reduce vector dimensions. Our method fundamentally differs from these schemes, as our proposal compresses the entries involved in secure aggregation while retaining the **intact** aggregation result of all original entries.

**Defense Against Malicious Server**. Several works indicate the malicious server can launch *Model Inconsistency Attacks* (Pasquini et al., 2022), *Multi-round Privacy Stealing Attacks* (So et al., 2023) and *Aggregation Falsification Attacks* (Guo et al., 2021). These studies also propose strategies to counter these attacks accordingly, only requiring minor modifications to the SAP process, as described in Section 3.4.

**Input constraints**. Several works (Bell et al., 2023; Lycklama et al., 2023) are proposed to mitigate *Poisoning Attacks* in FL. They delineate that the erroneous inputs of malicious users can result in the server obtaining an inaccurate global model, thereby harming the training task. They propose methodologies utilizing *Zero-Knowledge Proofs* to bound user inputs. However, their ability to prevent poisoning attacks is limited (Ma et al., 2023). Establishing strong constraints against malicious inputs remains an unresolved challenge, and it falls beyond the scope of this work. What's more, Mozaffari et al. (2023) propose *Federated Rank Learning* (FRL), where the server aggregates the parameter rankings instead of the model parameter updates. It can effectively resist poisoning attacks, and enable direct aggregations without any constraints on user submissions. Therefore, FRL can be combined with SAP, and we do not have to worry about whether PVF can be integrated with input constraints in this work.

## 3 PARTIAL VECTOR FREEZING

**Scenario.** In the $t$-th round of FL, the user set $\mathcal{U} = \{u_1, \ldots, u_n\}$ conduct local model training and submit model updates $\{\boldsymbol{x}^{i(t)}\}_{i \in \mathcal{U}} = \{(x_1^{i(t)}; \ldots; x_m^{i(t)})\}_{i \in \mathcal{U}}$ (*Original Vectors*) to the server $\mathcal{S}$. There might be $\eta$ ($\leq 30\%$) users that drop out during the aggregation due to network instability or other reasons and $\mathcal{U}'$ is the surviving user set. $\mathcal{S}$ aggregates the model updates to compute $\sum_{i \in \mathcal{U}'} \boldsymbol{x}^{i(t)}$ and redistributes the result to all users (*Plain Aggregation*). This iterative process continues until the completion of model training. SAPs can help obtain $\sum_{i \in \mathcal{U}'} \boldsymbol{x}^{i(t)}$ while ensuring the privacy of each individual $\boldsymbol{x}^{i(t)}$. Similar to other SAPs (Bonawitz et al., 2017; Stevens et al., 2022), we define the elements of $\boldsymbol{x}^{i(t)}$ ($i \in [1, n]$) within $\mathbb{Z}_p$ for some large public prime $p$ and assume there is a secure communication channel between each user and $\mathcal{S}$. In this section, our emphasis lies in introducing the computation methodology of PVF, specifically considering a single-round aggregation process with superscript "$^{(t)}$" omitted. To ensure multi-round privacy, employing a specialized user selection mechanism for each round is sufficient (Liu et al., 2023; So et al., 2023). For the summary of notations, please refer to Appendix A.

**Threat model**: Corrupt participants endeavor to infer the privacy of honest parties based on the messages they receive, i.e., the *Semi-honest Model*, and can fabricate messages, i.e., the *Active Adversary Model*. We assume malicious users do not exceed one-third of the total users, aligned with PracAgg (Bonawitz et al., 2017) and EffiAgg (Liu et al., 2022a). The lenient security assumptions of PVF allow its easy integration with secure aggregation protocols. When integrating with a SAP, the security assumptions originally employed in the corresponding protocol are adopted and we assume the integrated SAP can **reliably ensure the privacy of inputs**.

### 3.1 MOTIVATION

Within SAP, every minor operation on an entry accumulates $m$ times, ultimately imposing significant computational burdens on devices. For example, $u_i$ in PracAgg needs to perform the following calculations on each entry $x_j^i$ of $\boldsymbol{x}^i$ ($b$ and $s$ are user secret keys, PRG is a pseudorandom number generator):

$$y_j^i = x_j^i + \text{PRG}(b_i) + \sum_{h \in \mathcal{U}: i < h} \text{PRG}(s_{i,h}) - \sum_{h \in \mathcal{U}: i > h} \text{PRG}(s_{h,i}). \tag{1}$$

Based on these observations, we try to reduce the number of entries processed in secure aggregation while ensuring users receive all entries of the aggregated vector. To accomplish this objective, our

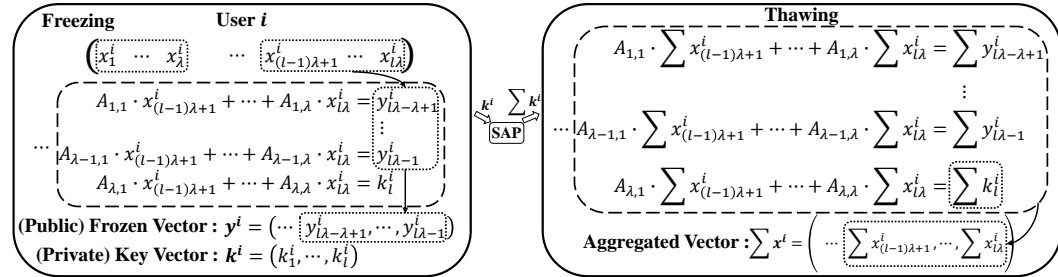

Figure 4: Workflow of $\lambda$-SecAgg with Main PVF.

mindset is to devise a module that can freeze certain entries (*Frozen Entries*) of the user's original vector, and only perform secure aggregation on the other entries (*SecAgg Entries*). Upon SAP completion, this module thaws the frozen entries, ensuring that their privacy is well protected throughout the entire process.

**Definition 1.** *Given an invertible $\lambda \times \lambda$ matrix $\boldsymbol{A}$ and a segment of original vector $\boldsymbol{x} = (x_1; \ldots; x_\lambda)$, we define Incomplete Matrix $\check{\boldsymbol{A}} = \boldsymbol{A}_{:\lambda-1,:}$ and Residual Vector $\boldsymbol{\alpha} = \boldsymbol{A}_{\lambda,:}$. The function for finding the solution set of a system of linear equations is:*

$$SLE_{AK}(\boldsymbol{A}\boldsymbol{x}) \to \boldsymbol{x}, \tag{2}$$

*where $AK$ denotes the additional knowledge. We use $\mathrm{rank}(\cdot)$ represents the rank of a matrix. Since $\mathrm{rank}(\boldsymbol{A}) = \mathrm{rank}(\boldsymbol{A}, \boldsymbol{A}\boldsymbol{x}) = \lambda$, the system of linear equations has a unique solution $\boldsymbol{x}$. However, due to $\mathrm{rank}(\check{\boldsymbol{A}}) = \mathrm{rank}(\check{\boldsymbol{A}}, \check{\boldsymbol{A}}\boldsymbol{x}) < \lambda$, the system has an infinitude of solutions (Suetin et al., 1989), also called an Under-determined System of Linear Equations.*

It can be seen that when $\check{\boldsymbol{A}}$ and $\check{\boldsymbol{A}}\boldsymbol{x}$ are known, in the absence of knowledge about $\boldsymbol{\alpha}\boldsymbol{x}$, $\boldsymbol{x}$ presents an infinite set of possibilities, rendering it impossible to determine that specific confidential vector. Motivated by this, we propose PVF.

## 3.2 MAIN METHOD

In this section, we present the computation process of the Main PVF module during a single aggregation round, depicted by the workflow shown in Figure 4.

**Phase 0: Main.Setup($\cdot$).** Generate an invertible matrix $\boldsymbol{A} \in \mathbb{Z}_p^{\lambda \times \lambda}$ randomly, and obtain $\boldsymbol{A}^{-1}$, $\check{\boldsymbol{A}}$, $\boldsymbol{\alpha}$, which are all **public** parameters. We refer to $\lambda$ as the *Compression Factor*.

**Phase 1: Main.Freeze($\cdot$).** For $i \in [1, n]$, randomly pad $\boldsymbol{x}^i$ to ensure the length of the padded vector is a multiple of $\lambda$. The padded vector is $\boldsymbol{x}_{pad}^i = (x_1^i; x_2^i; \ldots; x_{m'}^i)$, where $m' = l\lambda$. Then divide $\boldsymbol{x}_{pad}^i$ into $l$ groups: $\boldsymbol{x}_{pad}^i = (\boldsymbol{d}_1^i; \boldsymbol{d}_2^i; \ldots; \boldsymbol{d}_l^i)$, where $\boldsymbol{d}_j^i = (x_{(j-1)\lambda+1}^i; x_{(j-1)\lambda+2}^i; \ldots; x_{j\lambda}^i)$. Use the incomplete matrix $\check{\boldsymbol{A}}$ to compute *Frozen Vector*:

$$\begin{aligned} \boldsymbol{y}^i = (\boldsymbol{y}_1^i, \ldots, \boldsymbol{y}_l^i) &= \left( \left( y_1^i, \ldots, y_{\lambda-1}^i \right), \ldots, \left( y_{(l-1)\lambda+1}^i, \ldots, y_{(l-1)\lambda+\lambda-1}^i \right) \right) \\ &= \left( \check{\boldsymbol{A}}\boldsymbol{d}_1^i, \ldots, \check{\boldsymbol{A}}\boldsymbol{d}_l^i \right), \end{aligned} \tag{3}$$

and use the residual vector to compute *Key Vector*:

$$\boldsymbol{k}^i = \left( k_1^i, \ldots, k_l^i \right) = \left( \boldsymbol{\alpha}\boldsymbol{d}_1^i, \ldots, \boldsymbol{\alpha}\boldsymbol{d}_l^i \right). \tag{4}$$

**Phase 2: Main.SecAgg($\cdot$).** Users and $\mathcal{S}$ execute SAP, where the vector to be aggregated of user $i$ is $\boldsymbol{k}^i = (k_1^i, k_2^i, \ldots, k_l^i) \in \mathbb{Z}_p^l$. We require all users to send their respective $\boldsymbol{y}^i$ to $\mathcal{S}$ during SAP, thus **eliminating the need for additional interactions**. Upon completion of SAP, the surviving user set $\mathcal{U}'$ and $\mathcal{S}$ can receive the aggregated result of the key vectors: $\sum_{i \in \mathcal{U}'} \boldsymbol{k}^i$ or $Enc(\sum_{i \in \mathcal{U}'} \boldsymbol{k}^i)$ (in HE-based SAPs). Then $\mathcal{S}$ computes $\sum_{i \in \mathcal{U}'} \boldsymbol{y}^i$, immune to the impact of dropout users.

**Phase 3: Main.Thaw($\cdot$).** Thawing can be executed either at the server or user side, without compromising privacy and incurring any additional communication:

(1) *Thawing on the server side.* Based on the acquired $\sum_{i\in\mathcal{U}'} \boldsymbol{k}^i$ and $\sum_{i\in\mathcal{U}'} \boldsymbol{y}^i$, $\mathcal{S}$ can derive the aggregated result $\boldsymbol{sum}$. The correctness stems from the linearity of $\boldsymbol{A}$ and $\boldsymbol{\alpha}$:

$$
\begin{aligned}
\boldsymbol{z} &= \left( \left( \sum_{i\in\mathcal{U}'} \boldsymbol{y}_1^i, \sum_{i\in\mathcal{U}'} k_1^i \right), \ldots, \left( \sum_{i\in\mathcal{U}'} \boldsymbol{y}_l^i, \sum_{i\in\mathcal{U}'} k_l^i \right) \right) = \left( \left( \sum_{i\in\mathcal{U}'} \check{\boldsymbol{A}} \boldsymbol{d}_1^i, \sum_{i\in\mathcal{U}'} k_1^i \right), \ldots, \left( \sum_{i\in\mathcal{U}'} \check{\boldsymbol{A}} \boldsymbol{d}_l^i, \sum_{i\in\mathcal{U}'} k_l^i \right) \right) \\
&= \left( \sum_{i\in\mathcal{U}'} \boldsymbol{A}\boldsymbol{d}_1^i, \ldots, \sum_{i\in\mathcal{U}'} \boldsymbol{A}\boldsymbol{d}_l^i \right) = \left( \boldsymbol{A} \sum_{i\in\mathcal{U}'} \boldsymbol{d}_1^i, \ldots, \boldsymbol{A} \sum_{i\in\mathcal{U}'} \boldsymbol{d}_l^i \right)
\end{aligned}
\tag{5}
$$

Since $\boldsymbol{A}^{-1}$ and $\boldsymbol{z}$ are public, $\mathcal{S}$ can thaw frozen vectors and compute the aggregated results of all entries in the original vector by:

$$
\sum_{i\in\mathcal{U}'} \boldsymbol{d}_1^i = \boldsymbol{A}^{-1}\boldsymbol{z}_1 = \boldsymbol{A}^{-1}\boldsymbol{A} \sum_{i\in\mathcal{U}'} (x_1^i; x_2^i; \ldots; x_\lambda^i),
$$

$$
\vdots
\tag{6}
$$

$$
\sum_{i\in\mathcal{U}'} \boldsymbol{d}_l^i = \boldsymbol{A}^{-1}\boldsymbol{z}_l = \boldsymbol{A}^{-1}\boldsymbol{A} \sum_{i\in\mathcal{U}'} (x_{(l-1)\lambda+1}^i; x_{(l-1)\lambda+2}^i; \ldots; x_{l\lambda}^i).
$$

At this point, $\mathcal{S}$ completes the thawing phase and obtain:

$$
\boldsymbol{sum} = \left( \sum_{i\in\mathcal{U}'} \boldsymbol{d}_1^i; \ldots; \sum_{i\in\mathcal{U}'} \boldsymbol{d}_l^i \right) = \left( \sum_{i\in\mathcal{U}'} x_1^i; \ldots; \sum_{i\in\mathcal{U}'} x_{l\lambda}^i \right) = \sum_{i\in\mathcal{U}'} \boldsymbol{x}_{pad}^i.
\tag{7}
$$

Subsequently, it transmits $\boldsymbol{sum}$ to all online users. Upon receiving $\boldsymbol{sum}$, users can obtain the final aggregated result by removing the padding.

(2) *Thawing on the user side.* In certain SAPs (Aono et al., 2017; Xu et al., 2022), the aggregated results remain invisible to $\mathcal{S}$ to ensure the protection of users' intellectual property, among other goals. In such situations, $\mathcal{S}$ cannot perform the thawing. Instead, $\mathcal{S}$ sends $\sum_{i\in\mathcal{U}'} \boldsymbol{y}^i$ and $Enc(\sum_{i\in\mathcal{U}'} \boldsymbol{k}^i)$ to all surviving users. Users locally decrypt $Enc(\sum_{i\in\mathcal{U}'} \boldsymbol{k}^i)$ and perform the thawing process to obtain $\boldsymbol{sum}$. By removing the padding, users attain the final aggregated result.

### 3.3 DISRUPTING VARIABLES ELEMENT

While the server cannot obtain any individual element of $\boldsymbol{x}^i$ within Main PVF, it still obtains certain **linear relationships** involving the private entries. In this section, we present improvements to the Main PVF to ensure that the server cannot obtain any information about $\boldsymbol{x}^i$ from $\boldsymbol{y}^i$.

Firstly, we improve the process of generating $\boldsymbol{y}_j$ (ignoring the superscript "$i$" for clarity) in Equation (3) as follows:

$$
\begin{cases}
A_{1,1}(x_{(j-1)\lambda+1} + \underline{k_1 + \cdots + k_{\lfloor \frac{t}{\lambda} \rfloor}}) + \cdots + A_{1,\lambda}(x_{j\lambda} + \underline{k_{l-\lfloor \frac{t}{\lambda} \rfloor} + \cdots + k_l}) = y_{(j-1)\lambda+1} \\
\vdots \\
A_{\lambda-1,1}(x_{(j-1)\lambda+1} + \underline{k_1 + \cdots + k_{\lfloor \frac{t}{\lambda} \rfloor}}) + \cdots + A_{\lambda-1,\lambda}(x_{j\lambda} + \underline{k_{l-\lfloor \frac{t}{\lambda} \rfloor} + \cdots + k_l}) = y_{j\lambda}
\end{cases},
\tag{8}
$$

where $j \in [1, l]$ and $\boldsymbol{k}$ is the same as in Equation (4). Similar to Equation (5), upon thawing, the following can be derived:

$$
\begin{cases}
A_{1,1}(\sum x_{(j-1)\lambda+1}) + \cdots + A_{1,\lambda}(\sum x_{j\lambda}) = \sum y_{(j-1)\lambda+1} - \underline{\sum \sum_{o\in[1,\lambda]} A_{1,o} \sum_{r\in[(o-1)\lfloor \frac{t}{\lambda} \rfloor+1, o\lfloor \frac{t}{\lambda} \rfloor]} k_r} \\
\vdots \\
A_{\lambda-1,1}(\sum x_{(j-1)\lambda+1}) + \cdots + A_{\lambda-1,\lambda}(\sum x_{j\lambda}) = \sum y_{j\lambda-1} - \underline{\sum \sum_{o\in[1,\lambda]} A_{\lambda-1,o} \sum_{r\in[(o-1)\lfloor \frac{t}{\lambda} \rfloor+1, o\lfloor \frac{t}{\lambda} \rfloor]} k_r} \\
A_{\lambda,1}(\sum x_{(j-1)\lambda+1}) + \cdots + A_{\lambda,\lambda}(\sum x_{j\lambda}) = \sum k_j
\end{cases},
\tag{9}
$$

where the right side can be obtained given $\sum \boldsymbol{k}^i$ is known. Then, $\sum \boldsymbol{x}^i$ can be determined and the thawing phase is successfully completed. This improvement aims at **complicating the relationships among entries**. It can be seen that this additional operation only adds some vector-vector additions, which brings negligible computational overhead.

Secondly, similar to many hybrid schemes combining mask and DP (Bonawitz et al., 2017; Stevens et al., 2022; Liu et al., 2022c) (or encryption and DP (Wang et al., 2021)), DVE adds noise to $\boldsymbol{x}^i$ before aggregation to enhance privacy by:

$$
\boldsymbol{x}^i = \boldsymbol{x}^i + \boldsymbol{e},
\tag{10}
$$

where the Gaussian noise $e \sim N(0, \sigma^2)$. And $y^i = \check{A}(x^i + e) = \check{A}x^i + e'$ ($\check{A}$ is public). Therefore, given a uniformly random vector $w^i$, Lemma 3 in Appendix D.3 ensures that $(\check{A}, y^i)$ and $(\check{A}, w^i)$ are indistinguishable, which guarantees $\mathcal{S}$ **does not obtain private information from honest users through frozen vectors.** For clarity and conciseness, we do not differentiate the symbols of the original vector before and after adding noise.

### 3.4 Integrating PVF with Different SAPs

To resist **model inconsistency attacks**, appending the hash of the received model to the pseudo-random generator seed is sufficient, without incurring additional overhead (Ma et al., 2023). And utilizing an innovative user selection mechanism (So et al., 2023; Liu et al., 2023) is able to achieve **multi-round privacy**. To resist **aggregation falsification attacks**, verifiable protocols (Hahn et al., 2023) are able to verify the aggregation results through commitments sent by $\mathcal{S}$, and we provide *Result Verification Extension* in Appendix B.2 to enable PVF to integrate with such SAP. The pipeline of $\lambda$-SecAgg is shown in Figure 13. For the detailed portability analysis, please refer to Appendix C.

## 4 Security Analysis

Evidently, the information that adversaries can obtain about an honest participant only includes $sum$ and under-determined systems of linear equations ($y^i$). Randomly generating $A$ and performing certain **pre-checks** (as Section 4.1), the under-determined systems of linear equations can effectively preserve the privacy of each entry, which is also adopted by Liu et al. (2023). And in Section 4.2, we demonstrate adversaries cannot access $x^i$ throughout $\lambda$-SecAgg process.

### 4.1 Privacy of Each Element

In cases of improper selection of $A$, $\mathcal{S}$ can access the privacy of some specific elements within an original vector (if without DVE), as illustrated in Example D.1. In the implementation, we initially generate $A$ randomly, and we transform $\check{A}$ into *Reduced Row Echelon Form* and verify that no element can be deduced. This ensures privacy of every element. Unless specified otherwise, all subsequent references to $A$ in the following text are designed to guarantee each element privacy.

### 4.2 Security Analysis of Entire Vectors

Protocol security requires that adversaries can not obtain private information of any **individual** honest participant. The view of a participant consists of its internal information and received messages. Given a SAP that can maintain security in the active adversary setting, Theorem 1 guarantees its security when integrated with PVF. Theorem 1 demonstrates the information revealed during a real execution is **indistinguishable** from that obtained through a random simulation. The proof of Theorem 1 is is carried out in a *Random Oracle model*. In this model, we define a trapdoor function to inform $SIM$ of the **sum** of existing honest users' private information. During one execution of the protocol, $SIM$ can only access it once to obtain necessary information. Let $\mathcal{C}$ denote the set of malicious participants, which is a subset of $\mathcal{U} \cup \{\mathcal{S}\}$. The ideal function is defined as follows:

$$Ideal_{\{x^i\}_{i \in \mathcal{U} \setminus \mathcal{C}}}(L) = \begin{cases} \sum_{i \in L} x^i, \ L \subseteq (\mathcal{U} \setminus \mathcal{C}) \ and \ |L| > \left\lceil \frac{n}{3} \right\rceil \\ \bot, \ otherwise \end{cases}. \tag{11}$$

**Theorem 1** (Security against malicious participants). *Let $REAL_{\mathcal{C}}^{\mathcal{U}, \lambda}(\{x^i\}_{i \in \mathcal{U}}, \mathcal{U}')$ denote a random variable representing the joint view of adversaries in an actual protocol execution and $SIM_{\mathcal{C}}^{\mathcal{U}, \lambda}(\{x^i\}_{i \in \mathcal{U}}, \mathcal{U}')$ denote the joint view of adversaries in a simulated protocol execution. For all $\lambda > 2, \mathcal{U}, x_{i \in \mathcal{U}}^i, \mathcal{U}', \mathcal{C} \subseteq \mathcal{U} \cup \{\mathcal{S}\}$ and SAP that can ensure privacy in the active adversary setting, there exists a PPT simulator $SIM$ such that:*

$$SIM_{\mathcal{C}}^{\mathcal{U}, \lambda}(\{x^i\}_{i \in \mathcal{U}}, \mathcal{U}') \equiv REAL_{\mathcal{C}}^{\mathcal{U}, \lambda}(\{x^i\}_{i \in \mathcal{U}}, \mathcal{U}'), \tag{12}$$

*where "$\equiv$" denotes the distributions are identical.*

The proof of the theorem is provided in Appendix D.3.

## 5  Evaluation

### 5.1  Theoretical Complexity Analysis

**Communication.** PVF does not increase interaction-related overhead. The vectors sent from each user are $\boldsymbol{y}^i \in \mathbb{Z}_p^{(\lambda-1)\lceil \frac{m}{\lambda} \rceil}$ and $\boldsymbol{k}^i \in \mathbb{Z}_p^{\lceil \frac{m}{\lambda} \rceil}$, which contain the same number of entries as $\boldsymbol{x}_{pad}^i \in \mathbb{Z}_p^{m'}$. Hence, the theoretical communication complexity of SAP remains unchanged.

**Computation.** The additional computation operations required by PVF involve conducting $\lceil \frac{m}{\lambda} \rceil$ matrix-vector multiplications by both users and $\mathcal{S}$, incurring a computation cost of $O(\lambda m)$.

The theoretical complexity for one aggregation of users and $\mathcal{S}$, before and after PVF integration, is summarized in Table 4. For the method to **avoid padding**, please refer to Appendix E.4.

### 5.2  Experimental Settings

**Baselines**. We include 7 baselines that encompass 5 types of **masking (encryption) schemes**:

- *PPDL* (Aono et al., 2017), utilizing *Single-private-key HE* (**Type 1**), safeguards the global model, all while necessitating only a single interaction round between the users and $\mathcal{S}$.
- *EPPFL* (Li et al., 2022), relying on *Multi-private-key HE* (**Type 2**), requires two interaction round.
- *NIVP-DS* (Xu et al., 2022), based on *SMPC* (**Type 3**), requires two non-colluding servers, while involving only a single interaction round.
- *PracAgg* (Bonawitz et al., 2017), based on *Pair-wise Masking* (**Type 4**), is widely regarded as the state of the art, involving multiple interactions rounds.
- *PracAgg+* (Bell et al., 2020), a type-4 scheme, exhibits a more efficient communication structure.
- *EffiAgg* (Liu et al., 2022a), based on *Non-pair-wise Masking* (**Type 5**), requires server computation of discrete logarithms alongside multiple interactions between users and $\mathcal{S}$.
- *LPPFedL* (Wei et al., 2023), based on non-pair-wise masking, necessitates users to transmit multiple high-dimensional vectors along with multiple interactions.

To underscore the focal point of our work, our attention is exclusively directed towards secure-aggregation-related operations within the baselines, omitting other parts like "weight aggregation" in EPPFL. And masking schemes of other protocols that reduce **interaction-related** can be classified into the above five types, like Flamingo (Ma et al., 2023) (type 4), LTPA (Liu et al., 2023) (type 4).

**Experimental settings**. We run on a Linux workstation with 32GB of RAM and an AMD Ryzen 5 5600G. We use 1 NVIDIA GeForce RTX 3090 GPU only for model training, excluding the aggregation process. In all experimental settings, the space of the elements in the input vectors is 32-bit. Same with PracAgg, our main experiments are conducted under a semi-honest setting, with a specific focus on assessing the speed enhancement brought by PVF. We disregard operations such as digital signatures and public key infrastructure in the active adversary setting, which do not impact the asymptotics of the results (Bonawitz et al., 2017).

**Evaluation Metrics.** We use *Improvement Factor* $= \frac{\text{Time of (un)masking w/o PVF}}{\text{Time of (un)masking w/ PVF}}$ (speedup) to describe the efficacy of PVF. When evaluating communication overhead, we track costs for a single user, with the costs of $\mathcal{S}$ being $n$ times that of each user, a convention widely adopted in prior works (Liu et al., 2022a). The experimental results provided are the average outcomes from 5 repeated executions.

### 5.3  Performance Comparison

**Effectiveness of compressing costs.** Results in Table 1 indicate that integrating PVF can yield a speedup ranging from $\mathbf{70\times}$ to $\mathbf{99.5\times}$ for the majority of baselines when $\lambda = 100$. The inability to achieve a $\lambda\times$ speedup is attributed to interaction-related overhead within SAP such as key agreements and secret sharing. Overall, the gain from PVF is the weakest for LPPFedL. This is because LPPFedL increases user communication overhead to achieve highly lightweight masking and unmasking. Consequently, interaction-related overhead constitutes a more significant portion of its computation time. As the original vector length increases from $100k$ to $500k$, the proportion

Table 1: Comparison of secure aggregation protocols computation time (in milliseconds) and communication cost (in KB) before and after integrating PVF for a single round. The number of users $n$ is set to 100, with $\lambda$ set as 100. **w/** denotes the corresponding protocol integrated with PVF.

| Vector Length ($m$) | 100k | | | | | 500k | | | | |
|---|---|---|---|---|---|---|---|---|---|---|
| Operation Dropout rate ($\eta$) | Each user 0% | Server 0% | Server 10% | Server 30% | Comm. Cost 10% | Each user 0% | Server 0% | Server 10% | Server 30% | Comm. Cost 10% |
| PPDL | 139010 | 29759 | 27418 | 21789 | 27761 | 708579 | 149293 | 136155 | 107998 | 138679 |
| w/ PVF | 1401 | 304 | 283 | 225 | 859 | 7128 | 1571 | 1446 | 1133 | 4291 |
| Improvement Factor | **99.2×** | **97.9×** | **96.8×** | **96.8×** | **32.3×** | **99.4×** | **95.0×** | **94.1×** | **95.3×** | **32.3×** |
| EPPFL | 3464 | 755163 | 754261 | 749279 | 9962 | 17582 | 3763769 | 3732358 | 3745663 | 49798 |
| w/ PVF | 44 | 7686 | 7642 | 7711 | 681 | 219 | 38833 | 38924 | 39612 | 3400 |
| Improvement Factor | **78.7×** | **98.2×** | **98.7×** | **97.2×** | **14.6×** | **80.3×** | **97.0×** | **95.9×** | **94.6×** | **14.6×** |
| NIVP-DS | 15 | 386 | 372 | 331 | 586 | 72 | 1902 | 1834 | 1789 | 2932 |
| w/ PVF | 13 | 10 | 9 | 12 | 586 | 63 | 49 | 44 | 39 | 2931 |
| Improvement Factor | **1.2×** | **38.6×** | **41.3×** | **27.6×** | \ | **1.1×** | **38.9×** | **41.7×** | **45.9×** | \ |
| PracAgg | 13702 | 14249 | 139936 | 307031 | 785 | 73335 | 71607 | 697950 | 1515347 | 3910 |
| w/ PVF | 177 | 187 | 1472 | 3207 | 785 | 779 | 794 | 7063 | 15686 | 3910 |
| Improvement Factor | **77.4×** | **76.2×** | **95.1×** | **95.7×** | \ | **94.1×** | **90.2×** | **98.8×** | **96.6×** | \ |
| PracAgg+ | 2773 | 13973 | 39735 | 69623 | 782 | 14487 | 70359 | 197540 | 347055 | 3908 |
| w/ PVF | 38 | 159 | 410 | 724 | 782 | 179 | 787 | 2026 | 4319 | 3908 |
| Improvement Factor | **72.9×** | **87.9×** | **97.0×** | **96.2×** | \ | **80.9×** | **89.4×** | **97.5×** | **80.4×** | \ |
| EffiAgg | 1227 | 537771 | 537329 | 539814 | 783 | 6014 | 2662161 | 2730404 | 2637768 | 3908 |
| w/ PVF | 32 | 5677 | 5702 | 6135 | 783 | 99 | 27175 | 27440 | 28068 | 3908 |
| Improvement Factor | **38.3×** | **94.7×** | **94.2×** | **88.0×** | \ | **60.7×** | **98.0×** | **99.5×** | **94.0×** | \ |
| LPPFedL | 176 | 63 | 59 | 46 | 1564 | 846 | 319 | 291 | 218 | 7814 |
| w/ PVF | 22 | 21 | 20 | 17 | 1564 | 53 | 77 | 70 | 61 | 7814 |
| Improvement Factor | **8.0×** | **3.0×** | **3.0×** | **2.7×** | \ | **16.0×** | **4.1×** | **4.2×** | **3.6×** | \ |

of time spent on masking or encryption rises, so PVF exhibits a more noticeable acceleration effect for these SAPs in general. For PPDL and EPPFL (HE-based SAPs), only $\frac{1}{\lambda}$ of the original vector is transformed into **ciphertext**. Therefore, PVF brings them communication improvements of about $32.3\times$ and $14.6\times$, respectively.

**End-to-end comparison.** In Figure 2, the model employed is LeNet (LeCun et al., 1998), consisting of $61,706$ parameters. The dataset is MNIST (Deng, 2012) with each user possessing local data for only two labels. Across different aggregation methods, the model eventually achieves commendable training outcomes, and the integration of PVF showcases a significant acceleration effect, greatly reducing the training speed gap between using SAPs and plain aggregation. In Figure 3, consistent with FetchSGD (Rothchild et al., 2020), we use the ResNet9 (with 6.5M parameters), CIFAR10 dataset and PracAgg, with the optimal setting for FetchSGD and $k = 50,000$ for both FetchSGD and Top-k(Lu et al., 2023). It shows our method takes the **least** time and achieves the **best** accuracy because both FetchSGD and Top-k require more overhead for compression and PVF can ensure the intact aggregation of all entries in the original vector.

## 5.4 ABLATION STUDY

**$\lambda$.** We conduct experiments to analyze the variation in the acceleration effect of PVF on all baselines under different $\lambda$ values, as depicted in Figure 5. PVF exhibits a more pronounced acceleration effect for participants with substantial original computational overhead, such as PPDL users. For certain SAPs like EffiAgg, the improvement factor of $\mathcal{S}$ can reach up to $1000\times$. As $\lambda$ increases, the improvement factor of PVF gradually stabilizes and may experience a slight decline in certain SAPs. This trend emerges because when $\lambda$ becomes sufficiently large, the primary computational overhead in secure aggregation shifts from **masking-related** overhead to **interaction-related** overhead like secret sharing. It's worth noting that when $\lambda$ reaches a certain threshold, for certain entities with relatively low primary computational load, such as users in NIVP-DS and servers in LPPFedL, integrating PVF may increase their computational overhead (improvement factor less than 1). This occurs because the computation cost incurred by the linear transformation in PVF becomes notable. But the gains from PVF for the servers in NIVP-DS and users in LPPFedL remain enticing, rendering the cost of the PVF linear transformation negligible.

**Disrupting Variables Element.** We focus on the impact of DVE on the model's performance in this part. In the 32-bit input space, the $\sigma$ is set to 8783 ($> \frac{2 \times 1000}{\sqrt{2\pi}}$), which is equivalent to adding noise with a standard deviation of 0.0409 (consistent with (Stevens et al., 2022)) to the aggregation

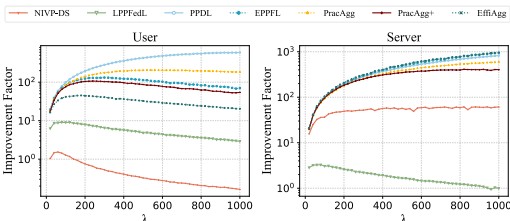

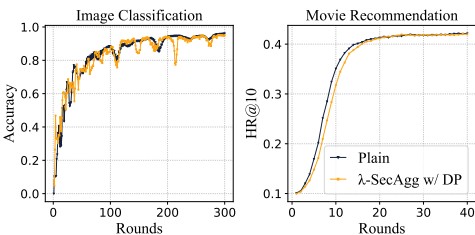

Figure 5: Computation acceleration for secure aggregation brought by different $\lambda$ in PVF. $n = 100$, $m = 100k$, and $\eta = 10\%$.

Figure 6: The impact of DVE on model performance in practical applications. In these two tasks, $n = 100$, $\eta = 100$, and $\lambda = 100$.

sum (float). And we explore the impact of DVE on model performance in two practical tasks: **(i) Image classification**, with LeNet model and MNIST dataset; **(ii) Movie recommendation**, using FedMF (Chai et al., 2020) model with the item embedding size of 32, i.e. 118,592 parameters in total, and ML-1M (Harper & Konstan, 2015) dataset. As shown in Figure 6, the impact of using PracAgg integrated with DVE on model performance is negligible.

$n$. Figure 7 displays the speedup effect of PVF across different $n$. The speedup remains consistently high for PracAgg+ and PPDL across various $n$. However, as $n$ rises, the speedup for PracAgg diminishes. This occurs because PracAgg necessitates multiple secret sharings for each user across the entire user set, along with multiple secret reconstructions by $\mathcal{S}$. These interaction-related overhead increases more significantly as $n$ increases. Nonetheless, even when $n$ reaches 1000, the gain brought by PVF for PracAgg servers remains above $\mathbf{85\times}$, with user gain still exceeding $\mathbf{55\times}$.

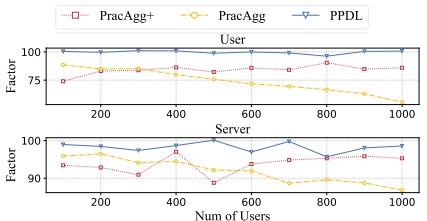

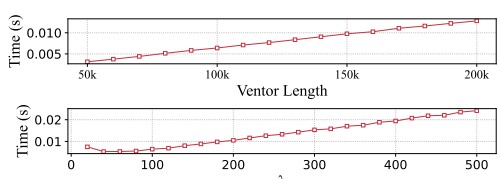

Figure 7: The gain introduced by PVF with respect to the number of users, with $\eta = 10\%$. $m = 100k$, and $\lambda = 100$.

Figure 8: Additional computational overhead incurred by PVF itself under different $\lambda$ and vector lengths.

**Transformation.** The additional overhead of integrating PVF primarily arises from Equations (3), (4) and (6). Figure 8 illustrates the **overall** additional computation costs of PVF concerning different $m$ and $\lambda$. It is evident that when $\lambda = 100$ and the original vector length is substantial (reaching $200k$), the computation time is less than 15 ms, nearly negligible compared to the computational time saved by PVF. As $\lambda$ gradually increases while vector length remains fixed, the number of loops during freezing and thawing computations decreases in the beginning because of the reduced number of groups. When $\lambda$ exceeds 60 and continues to increase, the scale of matrix-vector multiplication gradually rises, consequently increasing the computation cost. However, this cost remains consistently within the range of tens of milliseconds, posing a negligible computation burden.

## 6 CONCLUSION

We present a new perspective aimed at mitigating the formidable computation overhead of SAPs by reducing the number of involved entries while ensuring intact secure aggregation of original vectors. Based on this, we propose PVF, a concrete portable solution. After integrating with PVF, $\lambda$-SecAgg involves only $\frac{1}{\lambda}$ of original vectors. Moreover, we introduce the disrupting variables element to improve security. Extensive experiments showcase the remarkable improvements in acceleration and communication brought by PVF and its portability. Consequently, our method undoubtedly renders SAPs genuinely feasible, promising inspiration for future research endeavors.

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

## A  NOTATIONS

The primary notations used in this work are listed in Table 2.

Table 2: Notations.

| Symbol | Definition |
|---:|---|
| $n$ | Number of users |
| $m$ | Length of original vectors |
| $\eta$ | Dropout rate |
| $\lambda$ | Compression factor |
| $\mu$ | The number of broken elements within every $\lambda$ elements |
| $\mathcal{U}$ | The user set |
| $\mathcal{U}'$ | The surviving user set |
| $\mathcal{S}$ | The server |
| $\boldsymbol{x}^{i(t)}$ | Original vector of $i$ in the $t$-th round (with DP-based noise) |
| $\boldsymbol{A}$ | A $\lambda \times \lambda$ invertible matrix |
| $\boldsymbol{A}^{-1}$ | The inverse matrix of $\boldsymbol{A}$ |
| $\check{\boldsymbol{A}}$ | The matrix composed only of the first $\lambda - 1$ rows of $\boldsymbol{A}$ |
| $\boldsymbol{\alpha}$ | The vector composed of the $\lambda$-th row of $\boldsymbol{A}$ |
| $\check{\boldsymbol{A}}^{\mu+1}$ | $\check{\boldsymbol{A}}$ with $\mu$-security |
| $\boldsymbol{\alpha}^{\mu+1}$ | $\boldsymbol{\alpha}$ with $\mu$-security |
| $SLE_{AK}(\boldsymbol{Ax})$ | Function for computing $\boldsymbol{x}$ of $\boldsymbol{Ax}$ with the knowledge $AK$ |
| $\boldsymbol{x}_{pad}^{i(t)}$ | $\boldsymbol{x}^{i(t)}$ after padding |
| $m'$ | Length of $\boldsymbol{x}_{pad}^{i(t)}$ |
| $\boldsymbol{d}_j^{i(t)}$ | The $j$-th group of $\boldsymbol{x}_{pad}^{i(t)}$ |
| $\boldsymbol{y}^{i(t)}$ | Frozen vector of $\boldsymbol{x}_{pad}^{i(t)}$ |
| $\boldsymbol{GN}$ | DP-based noise generated by the Gaussian mechanism |
| $\boldsymbol{k}^{i(t)}$ | Key vector of $\boldsymbol{x}_{pad}^{i(t)}$ |
| $\boldsymbol{c}^{i(t)}$ | Commitment vector of $\boldsymbol{x}^{i(t)}$ |
| $\boldsymbol{\zeta}^{i(t)}$ | Corresponding random vector of $\boldsymbol{c}^{i(t)}$ |

## B  OPTIONAL EXTENSIONS

We introduce 3 optional extensions augmenting functionality and security for better portability: (i) *μ-security Extension*, which resists adversaries that possess prior knowledge of partial private

vectors; (ii) *Result Verification Extension* (RVE), which ensures the correctness of performing transformations to prevent malicious server-side computations, and (iii) *User Commitment Extension* (UCE), which ensures the uniqueness of users' original vectors to prevent malicious attempts at incorrect computations. The detailed pipeline for integrating PVF with all 4 extensions into SAP is depicted in Figure 13.

### B.1 $\mu$-SECURITY EXTENSION

**Threat**. Consider an extreme scenario where an extremely powerful malicious server $\mathcal{S}$ somehow obtains an element $x_e^i$ of $\boldsymbol{x}^i$, although this is generally unrealistic in most cases. In this case, $\mathcal{S}$, during the PVF (without DVE) execution, would acquire $\lambda - 1$ equations of the $j$-th group where $x_e^i$ lies, namely $\boldsymbol{y}_j^i = \check{\boldsymbol{A}} \boldsymbol{d}_j^i$ in Equation (3), along with $x_e^i$. This allows $\mathcal{S}$ to solve for all elements in the $j$-th group. Even though only a very small fraction of original vectors, $\frac{\lambda}{m}$, is leaked, we still aim to fortify against this. Assuming the adversary's capability is to steal $\mu$ ($< \lambda - 1$) elements from each group of $\boldsymbol{x}^i$.

We expand the definitions of incomplete matrix $\check{\boldsymbol{A}}$, and residual vector $\boldsymbol{\alpha}$ of PVF. We define *Incomplete Matrix with $\mu$-security* $\check{\boldsymbol{A}}^{\mu+1} = \boldsymbol{A}_{:\lambda-\mu-1,:}$ and *Residual Matrix with $\mu$-security* $\boldsymbol{\alpha}^{\mu+1} = \boldsymbol{A}_{\lambda-\mu:,:}$. Then *Frozen Vector with $\mu$-security* is:

$$\boldsymbol{y}^i = \left( \check{\boldsymbol{A}}^{\mu+1} \boldsymbol{d}_1^i, \check{\boldsymbol{A}}^{\mu+1} \boldsymbol{d}_2^i, \ldots, \check{\boldsymbol{A}}^{\mu+1} \boldsymbol{d}_l^i \right), \tag{13}$$

and *Key Vector with $\mu$-security* is:

$$\boldsymbol{k}^i = \left( \boldsymbol{\alpha}^{\mu+1} \boldsymbol{d}_1^i, \boldsymbol{\alpha}^{\mu+1} \boldsymbol{d}_2^i, \ldots, \boldsymbol{\alpha}^{\mu+1} \boldsymbol{d}_l^i \right). \tag{14}$$

Clearly, the aforementioned changes won't affect the execution process and security of PVF. However, the compression factor of PVF will decrease from $\lambda$ to $\frac{\lambda}{\mu+1}$ ($0 \leq \mu < \lambda - 1$). With these modifications, the adversary, apart from the known private elements, cannot obtain additional elements from the execution of PVF. Particularly, in the methodology outlined in Section 3.2, we consider $\mu = 0$.

### B.2 RESULT VERIFICATION EXTENSION

**Threat**. The malicious server might intentionally provide incorrect aggregated results to disrupt training, even if it **doesn't compromise user privacy**. When integrating PVF with SAPs featuring result verification capabilities, we contemplate incorporating RVE.

Most recent SAPs that can verify aggregated results typically necessitate honest and non-dropping participants, such as *Collectors* (Wang et al., 2023) and *Auxiliary Nodes* (Eltaras et al., 2023), introducing additional assumptions. Consequently, by utilizing this extension, we unavoidably add security assumptions. Without loss of generality, we adopt the approach and security assumptions from VerSA (Hahn et al., 2023) for PVF, which requires the server and users not to collude but does not need other trusted third parties. Most SAPs (Aono et al., 2017; Bonawitz et al., 2017; Bell et al., 2020; Li et al., 2022; Xu et al., 2022; Liu et al., 2022a; Stevens et al., 2022; Wei et al., 2023) lack the functionality to verify aggregated results. This extension merely constrains the server within the PVF module to obtain the correct $\sum_{i \in \mathcal{U}'} \boldsymbol{y}^i$, and it allows secure integration of PVF and SAPs in the face of aggregation falsification attacks.

**Phase 0: MainRV.Setup($\cdot$)**. Execute Main.Setup($\cdot$).

**Phase 1: MainRV.Freeze($\cdot$)**. Execute Main.Freeze($\cdot$).

**Phase 2: MainRV.SecAgg($\cdot$)**. During the execution of VerSA, all users obtain a set of random vectors $(\boldsymbol{\kappa}_1, \boldsymbol{\kappa}_2)$ derived from shared keys. In the verification step, users send $\grave{\boldsymbol{y}}^i = \boldsymbol{\kappa}_1 \boldsymbol{y}^i$ and $\acute{\boldsymbol{y}}^i = \boldsymbol{y}^i + \boldsymbol{\kappa}_2$ to the server, which simultaneously aggregates $\grave{\boldsymbol{y}}^i$ and $\acute{\boldsymbol{y}}^i$. In this process, the verification of the sum of frozen vectors and the sum of key vectors is combined, avoiding additional interaction. VerSA ensures users obtain a consistent $\mathcal{U}'$ and the correct $\sum_{i \in \mathcal{U}'} \boldsymbol{k}^i$. Users also receive $\sum_{i \in \mathcal{U}'} \grave{\boldsymbol{y}}^i$ and $\sum_{i \in \mathcal{U}'} \acute{\boldsymbol{y}}^i$.

**Phase 3: MainRV.Thaw($\cdot$)**. After receiving the results, users proceed with verification:

$$\sum_{i \in \mathcal{U}'} \grave{\boldsymbol{y}}^i \setminus \boldsymbol{\kappa}_1 \stackrel{?}{=} \sum_{i \in \mathcal{U}'} \acute{\boldsymbol{y}}^i - |\mathcal{U}'| \boldsymbol{\kappa}_2. \tag{15}$$

$y^i$ is unknown to $\mathcal{S}$ throughout the process. If the verification passes, execute Main.Thaw($\cdot$) for thawing on the user side. Otherwise, conclude that the server deviates from the protocol and terminate the execution.

### B.3 USER COMMITMENT EXTENSION

**Threat**. In PVF, if a malicious user $i$ uses inconsistent $\boldsymbol{d}_j^i$ and $\boldsymbol{d}_j'^i$ for freezing, resulting in obtaining $\check{\boldsymbol{A}}\boldsymbol{d}_j^i$ and $\boldsymbol{a}\boldsymbol{d}_j'^i$, or applies mismatched $\check{\boldsymbol{A}}$ and $\boldsymbol{\alpha}'$ to $\boldsymbol{d}_j^i$ for linear transformations, the frozen vector and key vector no longer correspond. Although this remains **harmless to the privacy** of honest users, it will lead to inaccuracies in the final aggregated result.

This malicious tampering is unrelated to SAPs. In the freezing phase of PVF, we use *Pedersen Commitment* (Pedersen, 1991) to ensure users' vector consistency. **Note** that, to enable verification, it's crucial to ensure that during the validation phase, the server is semi-honest and cannot collude with users. Otherwise, a malicious server could illegitimately approve malicious user actions, making user verification invalid, which is widely adopted in prior works (Rathee et al., 2023). Below, we outline UCE to modify the Main method to achieve the aforementioned functionality. UCE supports consistency verification of the user inputs in scenarios with advanced needs. Note that UCE is only applicable to SAPs where the server has access to the aggregated results.

**Phase 0: MainUC.Setup($\cdot$)**. Execute Main.Setup($\cdot$). Given the security parameter $\rho$, it generates the group $(\mathbb{G}_p, p, g)$, where $p$ is the order of $\mathbb{G}_p$ and $g$ is its generator. $h$ is an element of $\mathbb{G}_p$. $\mathbb{G}_p, p, g, h$ are public.

**Phase 1: MainUC.Freeze($\cdot$)**. The user $i \in [1, n]$ generates the *Random Vector* $\boldsymbol{\zeta}^i = (\zeta_1^i, \zeta_2^i, \ldots, \zeta_{l\lambda}^i)$ and calculates the *Commitment Vector*:

$$
\begin{aligned}
\boldsymbol{c}^i &= \left( \left( c_1^i, \ldots, c_\lambda^i \right), \ldots, \left( c_{(l-1)\lambda+1}^i, \ldots, c_{l\lambda}^i \right) \right) \\
&= ((g^{x_1^i} h^{\zeta_1^i}, g^{x_2^i} h^{\zeta_2^i}, \ldots, g^{x_\lambda^i} h^{\zeta_\lambda^i}), \ldots, (g^{x_{(l-1)\lambda+1}^i} h^{\zeta_{(l-1)\lambda+1}^i}, g^{x_{(l-1)\lambda+2}^i} h^{\zeta_{(l-1)\lambda+2}^i}, \ldots, g^{x_{l\lambda}^i} h^{\zeta_{l\lambda}^i})).
\end{aligned}
\tag{16}
$$

Execute Main.Freeze($\cdot$).

**Phase 2: MainUC.SecAgg($\cdot$)**. Execute Main.SecAgg($\cdot$), once completed, all users obtain the aggregated result $\boldsymbol{sum}$. Each user $i$ sends $\boldsymbol{c}^i$ and $\boldsymbol{\zeta}^i$ to the server. For $j \in [1, l]$, $r \in [1, \lambda]$, $\mathcal{S}$ validates:

$$
\prod_{i \in \mathcal{U}'} c_{(j-1)\lambda+r}^i \stackrel{?}{=} g^{sum_{(j-1)\lambda+r}} h^{\sum_{i \in \mathcal{U}'} \zeta_{(j-1)\lambda+r}^i}.
\tag{17}
$$

If the validation fails, it implies the user deviates from the protocol, and the protocol is terminated. Otherwise, it signifies that the user employs the consistent original vector when generating the frozen vector and key vector, and the utilized public parameters are accurate.

**Phase 3: MainUC.Thaw($\cdot$)**. Execute Main.Thaw($\cdot$).

Clearly, this extension will inevitably introduce a performance decrease. It's worth noting that in the majority of SAPs, many malicious user behaviors, such as incorrectly executing key agreements or submitting counterfeited secret shares, result in aggregation errors. And strictly enforcing user behavior remains a pending issue (Ma et al., 2023).

## C PORTABILITY ANALYSIS

PVF does not attempt to alter SAP, and the decoupling greatly enhances the portability. PPDL (Aono et al., 2017) and EPPFL (Li et al., 2022) employ homomorphic encryption, eliminating the need for secret sharing. NIVP-DS (Xu et al., 2022) constitutes a dual-server secure multi-party computation scheme, requiring users to share secrets between two servers. PracAgg (Bonawitz et al., 2017) and PracAgg+ (Bell et al., 2020) represent classic mask-based solutions, with PracAgg+ necessitating an additional user grouping process. EffiAgg (Liu et al., 2022a) and LPPFedL (Wei et al., 2023) respectively introduce specialized masking mechanisms to lightweight PracAgg. VerSA (Hahn et al., 2023) enables users to verify the aggregated result at the conclusion, and its integration with PVF requires RVE. LTPA (Liu et al., 2023) and MRSA (So et al., 2023) individually design specific user selection mechanisms to ensure privacy in **multi-round aggregation**. Resistance against **model**

Table 3: Symbolic representation of SAP execution process without and with PVF. The symbols' meanings are as follows: $US$: User Selection; $UG$: User Grouping; $KA$: Key Agreements among Users; $SS$: Secret Sharing; $E/M$: Encryption or Masking user's original vectors; $UA$: Upload and Aggregation; $D/U$: Decryption or Unmasking vectors; $Ver$: Verification of aggregated results; $F$: Freezing process in PVF Main Method, generating frozen vectors and key vectors; $T$: Thawing process in PVF Main Method, deriving all entries based on aggregated entries; $RV$: Result Verification Extension of PVF; $X'$: Specialized design for process $X$ in the corresponding protocol; $SA(\cdot)$: Operations involved in the aggregation process; $\square$: Protocol can execute in the semi-honest setting; $\blacksquare$: Protocol can execute in the active adversary setting.

| Type | Scheme | w/o PVF | w/ PVF |
|---|---|---|---|
| \ | PlainAgg | $[US, UA]$ | \ |
| HE-based | PPDL | $[US, \mathbf{SA}(KA, E/M, UA, D/U)]_\square$ | $[US, F, \mathbf{SA}(KA, E/M, UA, D/U), T]_\square$ |
| | EPPFL | $[US, \mathbf{SA}(KA, E/M, UA, D/U)]_\square$ | $[US, F, \mathbf{SA}(KA, E/M, UA, D/U), T]_\square$ |
| SMPC-based | NIVP-DS | $[US, \mathbf{SA}(KA, SS, E/M, UA, D/U)]_\square$ | $[US, F, \mathbf{SA}(KA, SS, E/M, UA, D/U), T]_\square$ |
| Mask-based | PracAgg | $[US, \mathbf{SA}(KA, SS, E/M, UA, D/U)]_{\square\blacksquare}$ | $[US, F, \mathbf{SA}(KA, SS, E/M, UA, D/U), T]_{\square\blacksquare}$ |
| | PracAgg+ | $[US, \mathbf{SA}(UG, KA, SS, E/M, UA, D/U)]_{\square\blacksquare}$ | $[US, F, \mathbf{SA}(UG, KA, SS, E/M, UA, D/U), T]_{\square\blacksquare}$ |
| | EffiAgg | $[US, \mathbf{SA}(KA, SS, E/M', UA, D/U')]_{\square\blacksquare}$ | $[US, F, \mathbf{SA}(KA, SS, E/M', UA, D/U'), T]_{\square\blacksquare}$ |
| | LPPFedL | $[US, \mathbf{SA}(KA, SS, E/M', UA, D/U')]_\square$ | $[US, F, \mathbf{SA}(KA, SS, E/M', UA, D/U'), T]_\square$ |
| Result Veri. | VerSA | $[US, \mathbf{SA}(KA, SS, E/M, UA, D/U, Ver)]_\square$ | $[US, F, \mathbf{SA}(KA, SS, E/M, UA, D/U, Ver), RV, T]_\square$ |
| Multi. Privacy | LTPA | $[US', \mathbf{SA}(KA, SS, E/M, UA, D/U)]_\square$ | $[US', F, \mathbf{SA}(KA, SS, E/M, UA, D/U), T]_\square$ |
| | MRSA | $[US', \mathbf{SA}(KA, SS, E/M, UA, D/U)]_\square$ | $[US', F, \mathbf{SA}(KA, SS, E/M, UA, D/U), T]_\square$ |
| Resist. M. Incon. | \ | $[US, \mathbf{SA}(KA, SS, E/M', UA, D/U)]_\square$ | $[US, F, \mathbf{SA}(KA, SS, E/M', UA, D/U), T]_\square$ |

**inconsistency attacks** can be achieved by making slight modifications to PRG without incurring additional overhead (Ma et al., 2023). PVF also fits the **asynchronous** setting. Since PVF itself is one-shot and decoupled from specific SAP, it does not affect the one-shot masking or recovery in asynchronous SAP (such as LightSecAgg (So et al., 2022)). We symbolically represent the entire process of federated learning aggregation in Table 3. It is evident that PVF is decoupled from SAPs, not interfering with the internal execution process of SAP.

# D  DETAILED SECURITY ANALYSIS

## D.1  EXAMPLE OF IMPROPER MATRIX

**Example 1.** *Consider a* $3 \times 3$ *matrix:*

$$\boldsymbol{A} = \begin{bmatrix} 1 & 2 & 3 \\ 1 & 3 & 3 \\ 1 & 2 & 4 \end{bmatrix}, \tag{18}$$

*which is an invertible matrix. The corresponding incomplete matrix is:*

$$\check{\boldsymbol{A}} = \begin{bmatrix} 1 & 2 & 3 \\ 1 & 3 & 3 \end{bmatrix}. \tag{19}$$

*Assume the original vector is* $\boldsymbol{x} = (x_1; x_2; x_3) = (1; 2; 3)$, *and the frozen vector is* $\check{\boldsymbol{A}}\boldsymbol{x} = (14, 16)$. $\mathcal{S}$ *obtains the under-determined system of linear equations as follows:*

$$\begin{cases} x_1 + 2x_2 + 3x_3 = 14 & (i) \\ x_1 + 3x_2 + 3x_3 = 16 & (ii) \end{cases} \tag{20}$$

While $\mathcal{S}$ cannot obtain the complete $\boldsymbol{x}$, it can deduce $x_2 = 2$ by $(ii) - (i)$. Unlike $\mathcal{S}$ obtaining prior knowledge of elements through attacks as discussed in Section B.1, here, $\mathcal{S}$ deduces $x_2 = 2$ through the computation process within PVF. However, since $\boldsymbol{A}$ is public, any maliciously constructed $\boldsymbol{A}$ can be easily detected by honest users.

### D.2 Supplementary Cryptographic Primitives

Here, we supplement the symmetric authenticated encryption and the digital signature we require in the active adversary model.

#### D.2.1 Symmetric Authenticated Encryption

Symmetric authenticated encryption can ensure the confidentiality of a message, including:

- $AE.gen(k) \rightarrow (sk)$, where $k$ is the security parameter. It outputs a secret key $sk$.
- $AE.enc(sk, m) \rightarrow (c)$. It encrypts the message $m$ using $sk$ and outputs the ciphertext $c$.
- $AE.dec(sk, c) \rightarrow m$ or $\perp$. If $sk$ is the correct key corresponding to the ciphertext $c$ and $c$ passes integrity verification, it outputs the plaintext $m$. Otherwise, it outputs an error symbol.

We need the encryption scheme to be indistinguishable under chosen plaintext attacks (IND-CPA) and ciphertext integrity (IND-CTXT) (Bellare & Namprempre, 2000). In Figure 13, we omit the encryption of messages before transmission and the decryption after reception by each participant. If any error occurs during encryption or decryption process, the protocol will be immediately terminated.

#### D.2.2 Digital Signature

Digital signature can ensure the authenticity and integrity of a message. We use the signature scheme that achieves security against universal forgery under chosen message attack (UF-CMA). The digital signature scheme consists of:

- $DS.gen(k) \rightarrow (sk, pk)$, where $k$ is the security parameter. It outputs a secret key $sk$ and a public key $pk$.
- $DS.sign(sk, m) \rightarrow (sig)$. It outputs a digital signature $sig$ on the message $m$.
- $DS.verify(sig, pk, m) \rightarrow True$ or $False$. It verifies whether the signature $sig$ is valid on $m$.

### D.3 Detailed explanation and proof of Theorem 1

First and foremost, it is evident that in PVF, the user only transmits $\mathbf{y}^i$ to the server, and the server only sends $\sum \mathbf{y}^i$ back to the users after SAP ends. Notably, $\mathbf{k}^i$ and $\sum \mathbf{k}^i$ are transmitted through the SAP, independent of PVF. In the active adversary model, we obviously cannot guarantee the correctness of the aggregation result because the malicious server can arbitrarily modify the result. However, we can guarantee the privacy of honest users' inputs. We provide a detailed explanation of the active attacks that malicious participants can launch within PVF and how PVF leverages the cryptographic primitives to defend against them.

- Forging fake users to participate in PVF. This type of attack, also known as a *Sybil Attack*, involves fake users reporting received information to the server. Such attacks primarily target scenarios where users share secret keys among themselves but keep the keys secret from the server, like PPDL. Alternatively, an attacker may attempt to forge a large number of fake users (more than $\frac{1}{3}|\mathcal{U}|$) to reconstruct users' private keys in the secret-sharing scheme. Since PVF does not involve information that is kept secret from the server but shared among all users, and consistent with the assumption in PracAgg that the number of malicious users does not exceed $\frac{1}{3}|\mathcal{U}|$, PVF is resistant to this type of attack.

- Attempting to forge or tamper with honest users' messages. Such attacks may occur in PVF in the following situations: malicious participants forging or tampering with an honest user's $\mathbf{y}^i$. This can be avoided by the digital signature $\sigma_1^i$ employed in PVF. Similarly, malicious participants may attempt to forge or tamper with $\sum \mathbf{y}^i$ sent by the server, which is prevented by the use of $\sigma_3$.

- Sending malformed messages. In PVF, such attacks include malicious users sending malformed ciphertexts of $\mathbf{y}^i$ or the malicious server sending malformed ciphertexts of $\sum \mathbf{y}^i$. Such attacks are prevented by the IND-CPA and IND-CTXT security of the symmetric authenticated encryption used in PVF. If decryption fails, the protocol is immediately terminated.

- Intercepting and stealing private information. Malicious adversaries may intercept messages sent by honest users to extract private information. This is effectively avoided by the symmetric authenticated encryption employed in PVF.

The use of symmetric authenticated encryption and digital signatures to ensure privacy under the active adversary model is a relatively mature application in the field of secure aggregation, and our design follows these existing works. Then we present the following lemmas:

**Lemma 1** (Privacy during the freezing phase). *Fix $p$, $\mathcal{U}$, $m$, $\lambda$, $\boldsymbol{A}$ and a private vector $\boldsymbol{x}^i = (\boldsymbol{d}_1^i, \dots, \boldsymbol{d}_{\lceil \frac{m}{\lambda} \rceil}^i)$ (with noise) of an honest user $i \in \mathcal{U}$. For any probabilistic polynomial-time (PPT) adversary $M$ who is given $\{\boldsymbol{y}^i\}_{i \in \mathcal{U}}$ and $\mathcal{C}$, the advantage of $M$ to obtain any unbroken element $a_j$ is defined as:*

$$Adv_M^y(\lambda) := Pr[SLE_\mathcal{C}(\check{\boldsymbol{A}}\boldsymbol{d}_j^i) \to a_j]_{j \in [1, \lceil \frac{m}{\lambda} \rceil]}. \tag{21}$$

*There exists a negligible function $\varepsilon$ such that $Adv_M^y(\lambda) \leq \varepsilon$.*

**Remark 1.** *The adversary can obtain $l(\lambda - 1)$ independent equations, with $l\lambda$ variables. Hence there are infinite solutions, and the probability of $M$ determining the unique $(\boldsymbol{x}^i)$ is $\frac{1}{\infty}$.*

**Lemma 2** (Privacy during the thawing phase). *Fix $p$, $\mathcal{U}$, $m$, $\lambda$, $\boldsymbol{A}$ and the sum of private vectors $\sum_{i \in \mathcal{U}'} \boldsymbol{x}^i$. For any PPT adversary $M$ who is given $\{\boldsymbol{y}^i\}_{i \in \mathcal{U}'}$, $\mathcal{C}$ and $\sum_{i \in \mathcal{U}'} \boldsymbol{k}^i$, the advantage of $M$ to obtain any unbroken element $a_j$ is defined as:*

$$Adv_M^{y,k}(\lambda) := Pr[SLE_\mathcal{C}(\boldsymbol{A} \sum_{j \in \mathcal{U}'} \boldsymbol{d}_j^i) \to a_j]_{j \in [1, \lceil \frac{m}{\lambda} \rceil]}. \tag{22}$$

*There exists a negligible function $\varepsilon$ such that $Adv_M^{y,k}(\lambda) \leq \varepsilon$.*

**Remark 2.** *$\mathcal{S}$ can obtain $\sum_{j \in \mathcal{U}'} \boldsymbol{d}_j^i$ by Equation (6) or Equation (9). For any individual $\boldsymbol{d}_j^i$ (with noise), the information known to $M$ is $\sum_{j \in \mathcal{U}'} \boldsymbol{d}_j^i$ and $\check{\boldsymbol{A}}\boldsymbol{d}_j^i$. So there are still infinite solutions, and $Adv_M^{y,k}(\lambda) = \frac{1}{\infty}$.*

**Lemma 3** (The hardness of the Learning With Errors decision problem). *Given a finite field $\mathbb{F}_q$ and a discrete probability distribution $\mathcal{X}$ over $\mathbb{F}_q$. Let $\boldsymbol{s} \in \mathbb{F}_q^v$ be a secret vector, $\boldsymbol{A} \in \mathbb{F}_q^{u \times v}$ be a matrix that is chosen uniformly at random and $\boldsymbol{e} \in \mathbb{F}_q^u$ be the error vector that is sampled from $\mathcal{X}$. $(v, q, \sigma)$ parameterize an LWE instance, where $\sigma$ is the standard deviation of $\mathcal{X}$. The Learning With Errors (LWE) (search) problem is to find $\boldsymbol{s}$, given the pair $(\boldsymbol{A}, \boldsymbol{b})$, where $\boldsymbol{b} = \boldsymbol{A}\boldsymbol{s} + \boldsymbol{e}$. And the LWE decision problem is to distinguish between two uniformly randomly generated pairs.*

**Remark 3.** *Regev (2009) shows that if the size of $q$ is polynomial in $v$ and $\mathcal{X}$ is a discrete Gaussian distribution on $\mathbb{F}_q$ with standard deviation $\sigma > \frac{2\sqrt{v}}{\sqrt{2\pi}}$, the LWE decision problem is at least as hard as the LWE search problem and solving the LWE search problem can be reduced to solving the Shortest Vector Problem. In DVE, $v = \lambda$, and we use $\mathbb{Z}_p$ as $\mathbb{F}_q$.*

We use a standard hybrid argument to prove the theorem.

*Proof.* We define a sequence of hybrid distributions $H_0, H_1, \dots$ to denote a series of modifications to $REAL$, which can finally get $SIM$. We prove $SIM$ and $REAL$ are indistinguishable by proving two adjacent hybrids are indistinguishable.

$H_0$ In this hybrid, $SIM$ is exactly the same as $REAL$.

$H_1$ This hybrid is distributed similarly to the previous one, except for the following modifications. $SIM$ obtains $\sum_{i \in \mathcal{U}' \backslash \mathcal{C}} \boldsymbol{x}^i$ by calling $Ideal_{\{\boldsymbol{x}^i\}_{i \in \mathcal{U} \backslash \mathcal{C}}}(\mathcal{U}' \backslash \mathcal{C})$. $SIM$ aborts if there is an illegal request. We replace the ciphertexts of $\{\boldsymbol{y}^i\}_{i \in \mathcal{U}}$ with the ciphertexts of uniformly random vectors $\{\boldsymbol{w}^i\}_{i \in \mathcal{U}}$ that satisfy $\sum_{i \in \mathcal{U}' \backslash \mathcal{C}} \boldsymbol{w}^i = \sum_{i \in \mathcal{U}' \backslash \mathcal{C}} \boldsymbol{y}^i$. $\sum_{i \in \mathcal{U}' \backslash \mathcal{C}} \boldsymbol{y}^i$ can be can be computed from Equation (9) based on $\sum_{i \in \mathcal{U}' \backslash \mathcal{C}} \boldsymbol{x}^i$. The IND-CPA and IND-CTXT security of symmetric authenticated encryption guarantees the distribution of this hybrid is indistinguishable from the previous one.

$H_2$ This hybrid is distributed exactly as the previous one, except $SIM$ aborts if there is an invalid signature ($\sigma_1^i$, $\sigma_2^i$ or $\sigma_3$). The UF-CMA security of the digital signature scheme can ensure $\mathcal{C}$ cannot

forge any valid signature of an honest user, so the distribution of this hybrid is indistinguishable from the previous one.

$H_3$ This hybrid is distributed similarly to the previous one, except for the following modifications. **Firstly**, according to the security analysis process of the integrated SAP, we replace the corresponding messages conveyed in SAP with random strings of equal length. **Secondly**, we replace the frozen vectors $\{\boldsymbol{y}^i\}_{i \in \mathcal{U}}$ received by $\mathcal{S}$ with uniformly random vectors $\{\boldsymbol{w}^i\}_{i \in \mathcal{U}}$. $\boldsymbol{x}^i$ is added with noise $(\check{\boldsymbol{A}}\boldsymbol{e})$ through Equation (10). Therefore, Lemma 3 ensures that $(\check{\boldsymbol{A}}, \boldsymbol{y}^i)$ and $(\check{\boldsymbol{A}}, \boldsymbol{w}^i)$ are indistinguishable, which guarantees $\mathcal{S}$ does not obtain private information from honest users through frozen vectors. Therefore, the security of SAP and the lemmas ensure the distribution of this hybrid is indistinguishable from the previous one.

Therefore, the distribution of $SIM$ which is the same as $H_3$ is indistinguishable from $REAL$. $SIM$ does not depend on the inputs of honest parties, and $\mathcal{C}$ can only learn about the sum of original vectors. If too many users drop out, SAP will abort and still guarantee the above conclusion. Clearly, the security of $\lambda$-SecAgg still holds in the semi-honest setting. $\square$

# E ADDITIONAL COMPARISON AND EXPERIMENT

## E.1 THEORETICAL COMPLEXITY COMPARISON

Table 4 indicates for some SAPs, the theoretical computation complexity increases after integrating PVF. This is attributed to the transformation required for the entire original vector within PVF. However, intuitively, the computation time for secure aggregation per entry (e.g., homomorphic encryption, PRG expansions, or modular exponentiations) tends to be **significantly greater compared to the computation time per entry in linear transformations**. This suggests that PVF still manages to compress computation overhead, which is evident in the experiment outcomes.

In practice, the choice of $\lambda$ mainly considers: (i) Security requirements in DVE (see Lemma 3). (ii) SAP. For schemes with more masking-related overhead, a larger $\lambda$ performs better. For schemes with more interaction-related overhead, a smaller $\lambda$ performs better (as in Section 5.3). (iii) $m$. For larger $m$, masking-related overhead is greater, so a larger $\lambda$ performs better.

Table 4: Theoretical complexity of SAP without and with PVF of single-round aggregation in the semi-honest setting. $\boldsymbol{O}(\cdot)(\cdot)$ in the rightmost column indicates the communication complexity pertains to users, followed by the number of interactions between users and the server. "P. of G. M." stands for privacy of the global model. In the real world, $p \gg m \gg n$. $\uparrow$ indicates the integration of PVF into the protocol would increase its theoretical complexity.

| SAP | Each user | | Server | | P. of G. M. | Communi. (Inter.) |
|---|---|---|---|---|---|---|
| | w/o PVF | w/ PVF | w/o PVF | w/ PVF | | |
| PPDL | $O(m)$ | $O(\lambda m)\uparrow$ | $O(mn)$ | $O(\frac{1}{\lambda}mn + \lambda m)$ | ✓ | $O(m)$ (1) |
| EPPFL | $O(m)$ | $O(\lambda m)\uparrow$ | $O(mn)$ | $O(\frac{1}{\lambda}mn + \lambda m)$ | × | $O(m)$ (2) |
| NIVP-DS | $O(m)$ | $O(\lambda m)\uparrow$ | $O(mn)$ | $O(\frac{1}{\lambda}mn + \lambda m)$ | ✓ | $O(m)$ (1) |
| PracAgg | $O(mn + n^2)$ | $O(\frac{1}{\lambda}mn + \lambda m + n^2)$ | $O(mn^2)$ | $O(\frac{1}{\lambda}mn^2 + \lambda m)$ | × | $O(m + n)$ (4) |
| PracAgg+ | $O(mlogn + log^2n)$ | $O(\frac{1}{\lambda}mlogn + \lambda m + log^2n)$ | $O(mnlogn + nlog^2n)$ | $O(\frac{1}{\lambda}mnlogn + \lambda m + nlog^2n)$ | × | $O(m + logn)$ (4) |
| EffiAgg | $O(m + n^2)$ | $O(\lambda m + n^2)\uparrow$ | $O(m\sqrt{p} + n)$ | $O(\frac{1}{\lambda}m\sqrt{p} + \lambda m + n)$ | × | $O(m + n)$ (4) |
| LPPFedL | $O(m + n^2)$ | $O(\lambda m + n^2)\uparrow$ | $O(m + n)$ | $O(\lambda m + n)\uparrow$ | × | $O(m + n)$ (4) |

## E.2 MORE IMPLEMENTATION DETAILS

We implement the baselines using Python. Specifically, we utilize *AES-GCM* with 128-bit keys for the symmetric authenticated encryption, standard $(t, n)$ *Shamir Secret Sharing* (Shamir, 1979), *AES* in counter mode for the pseudorandom generator, *SHA-256* hash to implement a homomorphic pseudorandom generator for EffiAgg, *Sympy* library (Meurer et al., 2017) to compute discrete logarithms for EffiAgg, and *Paillier Encryption* with 1024-bit keys for PPDL. In Section 3.3, experimental settings of the image classification task is the same as Figure 2, and for the movie recommendation task, the dataset is split using a *Leave-One-Out* (He et al., 2017) approach for training and testing, where users with fewer than 10 records are excluded, using *Hit Ratio* (HR) as metrics to assess the performance of the recommendations, with higher values indicating superior effectiveness.

In Figure 2, we use three widely discussed SAPs: PracAgg+, PracAgg, and PPDL. NIVP-DS falls behind due to the requirement of two non-colluding servers, while LPPFedL demands increased communication overhead. Remarkably, PPDL, previously considered impractical due to its use of HE, has its computational overhead significantly mitigated by integrating PVF. And its advantages of operating with a single server, single-interaction communication, and preserving the global model make it more appealing.

## E.3 FOR FL OF LLM

LLMs have a profound impact on the entire AI research community due to their excellent contextual learning and instruction following ability (Zhao et al., 2023). Given privacy concerns, training (or fine-tuning) LLM in a federated setting has been explored (Hilmkil et al., 2021; Ye et al., 2024). In the context of LLMs, during the aggregation process, the length of user original vectors reaches billions. Taking Llama2-7B (Touvron et al., 2023) as an example, assuming 1% of the parameters need to be updated during the fine-tuning, which is 700M, the computational cost of using a general secure aggregation scheme is unimaginable, making PVF particularly important. As shown in Table 5, the time required to train one round without using PVF can basically meet the requirement of training 100 rounds with PVF. At present, FL for LLM among a multitude of lightweight clients is unrealistic. The SOTA scheme OpenFedLLM (Ye et al., 2024) involves only $n \in \{2, 4, 5\}$ clients per round (using Llama2-7B). And PVF can contribute to future FedLLMs and involving more clients.

Table 5: Estimated computational overhead per round with and without PVF for fine-tuning Llama2-7B. $n = 100$, $m = 700M$, $\lambda = 100$, and $\eta = 0\%$.

| Scheme | Each user | | Server | |
|--------|-----------|-----------|-----------|-----------|
| | w/o | w/ | w/o | w/ |
| PPDL | ~300h | ~3h | ~60h | ~0.6h |
| PracAgg | ~27h | ~0.27h | ~27h | ~0.27h |
| PracAgg+ | ~5h | ~0.05h | ~27h | ~0.27h |

## E.4 THE IMPACT OF PADDING

PVF necessitates padding the original vector, thereby increasing the vector size. In theory, the additional cost that padding introduces in computation and communication does not exceed $\frac{\lambda}{m}$ of the original, as **the maximum value of padding length** is $\lambda$. Table 6 showcases the impact of no padding versus padding $\lambda$ entries on the computation and communication overhead, where $\frac{\lambda}{m} = 0.001$. It's evident that the overhead induced by padding is almost negligible. To strictly adhere to the principle of "**not increasing any communication overhead**", we present a method to avoid padding. We extract the first $\lfloor \frac{m}{\lambda} \rfloor \lambda$ entries from the original vector and apply PVF to them. The remaining entries, which are fewer than $\lambda$, are appended to the key vector $\boldsymbol{k}$ for participation in the secure aggregation. This approach allows us to obtain the aggregate result of the entire original vector while eliminating the need for padding.

Table 6: Comparison of overhead with and without padding. $n = 100$, $m = 100k$, $\lambda = 100$, and $\eta = 10\%$.

| Scheme | User comp. (ms) | | Server comp. (ms) | | Comm. cost (KB) | |
|--------|-------|-----------|--------|-----------|--------|-----------|
| | No pad | Pad | No pad | Pad | No pad | Pad |
| PPDL | 1402 | 1403, ↑1 | 303 | 304, ↑1 | 859 | 860, ↑1 |
| EPPFL | 41 | 42, ↑1 | 7337 | 7351, ↑14 | 681 | 681, ↑0 |
| NIVP-DS | 13 | 13, ↑0 | 10 | 10, ↑0 | 587 | 587, ↑0 |
| PracAgg | 162 | 164, ↑2 | 1456 | 1463, ↑7 | 785 | 785, ↑0 |
| PracAgg+ | 38 | 39, ↑1 | 405 | 409, ↑4 | 783 | 783, ↑0 |
| EffiAgg | 29 | 32, ↑3 | 5404 | 5482, ↑78 | 783 | 783, ↑0 |
| LPPFedL | 19 | 19, ↑0 | 20 | 20, ↑0 | 1564 | 1564, ↑0 |

### E.5 Memory Usage

In Figure 9, we evaluate the memory usage of each user and $\mathcal{S}$ during PracAgg without PVF ($\lambda = 0$) and with PVF ($\lambda = 100, 1000$).

**For each user**, the increased memory usage for $\lambda = 1000$ compared to $\lambda = 100$ is due to the need to store a larger transformation matrix ($\lambda \times \lambda$) required by PVF. The increase in memory usage for $\lambda = 0$ compared to $\lambda = 100$ is because PVF reduces the number of random numbers generated ($nm \to \frac{nm}{\lambda}$) and decreases the scale of vector addition computations ($m \to \frac{m}{\lambda}$) during the masking process.

**For $\mathcal{S}$**, all three methods require summing up the vectors uploaded by all users, with a memory usage of approximately $O(mn)$, so the additional $O(\lambda^2)$ overhead introduced by PVF is negligible. The increased memory usage for $\lambda = 0$ compared to $\lambda = 100$ is due to PVF reducing the number of random numbers generated ($(\eta(1-\eta)n^2 + (1-\eta)n)m \to \frac{(\eta(1-\eta)n^2+(1-\eta)n)m}{\lambda}$) and decreasing the scale of vector addition computations during the unmasking process.

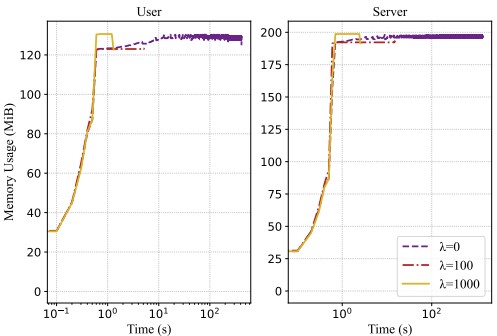

Figure 9: Memory usage with different $\lambda$. $n = 100$, $m = 100k$, and $\eta = 10\%$.

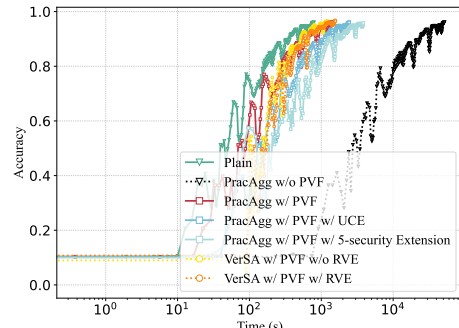

Figure 10: Comparison of various aggregation methods, with $n = 100, \eta = 5\%, \lambda = 100$.

### E.6 Evaluation of extensions

**End-to-end comparison.** In accordance with the experimental settings of Figure 2, we evaluate the impact of RVE, UCE, and $\mu$-security on the overall model training time (as illustrated in the Figure 10). The acceleration effect and communication expansion when integrating different extensions are 34.5× and 1.4× (RVE), 21.1× and 2.9× (UCE), 13.5× and 1.0× (5-security).

**$(\boldsymbol{\lambda}, \boldsymbol{\mu})$.** We evaluate the speedup with different $(\lambda, \mu) \in \{100, 300, 500, 700, 1000\} \times [1, 10]$. The integration of PVF with the $\mu$-security extension reduces the entries of vectors involved in secure aggregation to $\frac{\mu+1}{\lambda}(\mu < \lambda - 1)$ of their original size. Figure 11 showcases incorporating the $\mu$-security extension does diminish the improvement factor. However, even when $\mu = 0.1\lambda$, PVF still yields an acceleration gain of 10× along with communication improvements exceeding 5×.

**RVE and UCE.** UCE necessitates users to commit to each dimension of the original vectors, while $\mathcal{S}$ needs to validate each dimension. RVE requires users to submit frozen vectors twice and $\mathcal{S}$ to perform summations twice. Thereby, both UCE and RVE impose on each participant an additional communication complexity of $O(m)$ and computation complexity of $O(m)$.

Figure 12 presents the overhead required for SAPs with different extensions. Compared to not integrating PVF, the computation overhead after adding extensions is still nearly **an order of magnitude lower**. Communication costs from extensions primarily stem from the transmission of additional vectors, such as commitment vectors. We leave mitigating these overheads to future work.

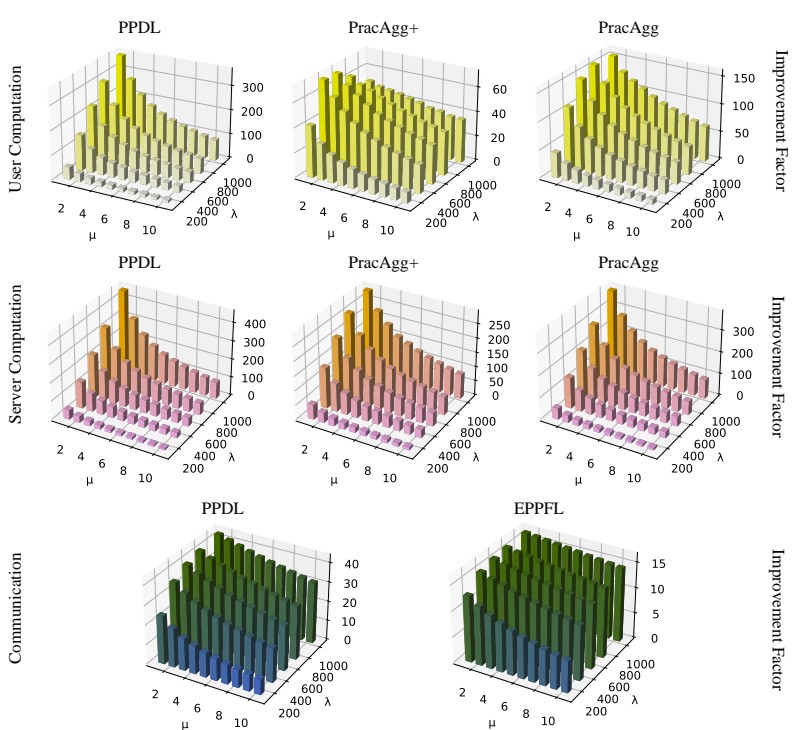

Figure 11: The impact of different $(\lambda, \mu)$ on improvement factor. $n = 100$, $m = 100k$, $\lambda = 100$, and $\eta = 10\%$.

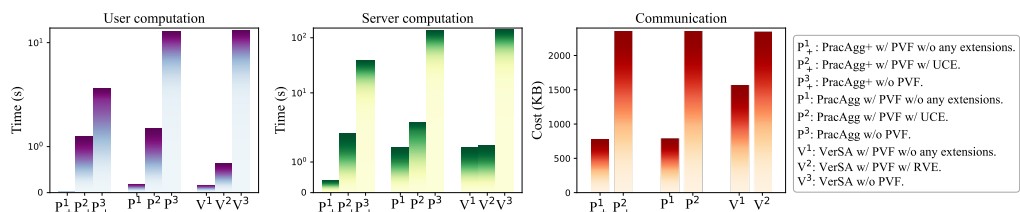

Figure 12: Comparison of costs across different extensions. $n = 100$, $m = 100k$, $\lambda = 100$, and $\eta = 10\%$.

**Participants**: $\mathcal{S}$ and User set $\mathcal{U} = \{u_1, u_2, \ldots, u_n\}$.
**Public Inputs**: $\boldsymbol{A}$, $\mu$, $\hat{\boldsymbol{A}}^{\mu+1}$, $\boldsymbol{\alpha}^{\mu+1}$, $\lambda$, $\mathbb{Z}_p$, $g$ and $h$. Users' public keys for signatures $\{sig_i^{pk}\}_{i\in\mathcal{U}}$ and the server's public key for signatures $sig_S^{pk}$.
**Private Inputs**: Original vectors $\{\boldsymbol{x}^{i(t)}\}_{i\in\mathcal{U}}$ of $t$-th iteration. Users' secret keys for signatures $\{sig_i^{sk}\}_{i\in\mathcal{U}}$ and the server's secret key for signatures $sig_S^{sk}$.
**Outputs**: Surviving user set $\mathcal{U}'$, $\sum_{i\in\mathcal{U}'}\boldsymbol{x}^{i(t)}$.
• **Phase 1 Freezing**
  User $i \in \mathcal{U}$:
    - pad $\boldsymbol{x}^{i(t)}$ randomly and group the entries.
    - add noise to $\boldsymbol{x}^{i(t)}$ via Equation (10).
    - calculate key vector $\boldsymbol{k}^{i(t)}$ via Equation (14).
    - calculate frozen vector $\boldsymbol{y}^{i(t)}$ via Equation (8) and Equation (13).
    - generate random vector $\boldsymbol{\zeta}^{i(t)} = (\zeta_1^{i(t)}, \zeta_2^{i(t)}, \ldots, \zeta_{l\lambda}^{i(t)})$ and calculate commitment vector $\boldsymbol{c}^{i(t)}$ via Equation (16).
    - obtain $m_1^i = \boldsymbol{y}^{i(t)}$(Do not send $\boldsymbol{y}^{i(t)}$ when there is RVE)$||\boldsymbol{\zeta}^{i(t)}||\boldsymbol{c}^{i(t)}$, send $\sigma_1^i \to DS.sign(sig_i^{sk}, m_1^i)$ to $\mathcal{S}$.
• **Phase 2 SecAgg**
  $\mathcal{S}$ and Users:
    - execute **SAP** for $\{\boldsymbol{k}^{i(t)}\}_{i\in\mathcal{U}}$.
        * $\cdots$
        * users get $(\boldsymbol{\kappa_1}, \boldsymbol{\kappa_2})$ and obtain $m_2^i = \check{\boldsymbol{y}}^{i(t)}(\boldsymbol{\kappa_1}\boldsymbol{y}^{i(t)})||\hat{\boldsymbol{y}}^{i(t)}(\boldsymbol{y}^{i(t)} + \boldsymbol{\kappa_2})$, send $\sigma_2^i \to DS.sign(sig_i^{sk}, m_2^i)$ to $\mathcal{S}$.
        * $\cdots$
    - all participants receive $\mathcal{U}'$ and $\sum_{i\in\mathcal{U}'}\boldsymbol{k}^{i(t)}$ (or $Enc(\sum_{i\in\mathcal{U}'}\boldsymbol{k}^{i(t)})$).
  $\mathcal{S}$:
    - if $DS.verify(\sigma_1^i, sig_i^{pk}, m_1^i) \to False$, abort. Otherwise, calculate $\sum_{i\in\mathcal{U}'}\boldsymbol{y}^{i(t)}$(can not get $\sum_{i\in\mathcal{U}'}\boldsymbol{y}^{i(t)}$) calculate $\sum_{i\in\mathcal{U}'}\check{\boldsymbol{y}}^{i(t)}$ and $\sum_{i\in\mathcal{U}'}\hat{\boldsymbol{y}}^{i(t)}$.
    - for $j \in [1, l]$, $r \in [1, \lambda]$, reveal the commitments via Equation (17). If verification fails, abort.
• **Phase 3 Thawing**
  *Thawing on the server side*
  $\mathcal{S}$:
    - calculate $\boldsymbol{sum} = \sum_{i\in\mathcal{U}'}\boldsymbol{x}_{pad}^{i(t)}$ via Equation (9) and send $\boldsymbol{sum}$ and $\sigma_3 \to DS.sign(sig_S^{sk}, \boldsymbol{sum})$ to $i \in \mathcal{U}'$.
  User $i \in \mathcal{U}'$:
    - receive $\sum_{i\in\mathcal{U}'}\boldsymbol{x}^{i(t)}$, if $DS.verify(\sigma_3, sig_S^{pk}, \boldsymbol{sum}) \to False$, abort. Otherwise, unpad and output.
  *Thawing on the user side*
  $\mathcal{S}$:
    - send $\sum_{i\in\mathcal{U}'}\boldsymbol{y}^{i(t)}$, $Enc(\sum_{i\in\mathcal{U}'}\boldsymbol{k}^{i(t)})$ and $\sigma_3 \to DS.sign(sig_S^{sk}, \sum_{i\in\mathcal{U}'}\boldsymbol{y}^{i(t)}||Enc(\sum_{i\in\mathcal{U}'}\boldsymbol{k}^{i(t)}))$ to $i \in \mathcal{U}'$.
  User $i \in \mathcal{U}'$:
    - if $DS.verify(\sigma_3, sig_S^{pk}, \sum_{i\in\mathcal{U}'}\boldsymbol{y}^{i(t)}||Enc(\sum_{i\in\mathcal{U}'}\boldsymbol{k}^{i(t)})) \to False$, abort. Otherwise, verify $\sum_{i\in\mathcal{U}'}\boldsymbol{y}^{i(t)}$ via Equation (15). If verification fails, abort.
    - decrypt $Enc(\sum_{i\in\mathcal{U}'}\boldsymbol{k}^{i(t)})$.
    - calculate $\boldsymbol{sum} = \sum_{i\in\mathcal{U}'}\boldsymbol{x}_{pad}^{i(t)}$ via Equation (9), unpad and output.

Figure 13: The pipline of $\lambda$-SecAgg with all 4 extensions for one aggregation. The red and underlined parts are required in the user commitment extension. The blue and underlined parts are required in the result verification extension.

