# OpenReview forum: "$\lambda$-SecAgg: Partial Vector Freezing for Lightweight Secure Aggregation in Federated Learning"
_ICLR.cc/2025/Conference — Submitted to ICLR 2025_

### Official Review · Reviewer_r5Dg · 2024-10-31

**Soundness:** 1
**Presentation:** 2
**Contribution:** 1
**Rating:** 1
**Confidence:** 4

**Summary:**

The paper studies how to reduce the computational cost in privacy-preserving federated learning (FL) with secure aggregation (SecAgg). SecAgg is a primitive that improves the privacy-utility trade-off in FL as it hides individual model updates sent to the server. However, most efficient applications require to mask each parameter of the model with random noise, incurring into a large computational cost if models are big.

The current work proposes a technique that reduces the computational cost by only performing SecAgg to a subset of model parameters, while still recovering the (claimed to be) private aggregation of the entire model.

**Strengths:**

- Reducing the communication cost in privacy preserving ML an interesting topic.

- The presentation of the protocol is fairly clear.

**Weaknesses:**

# Main Weaknesses

The major weakness of the protocol is the **lack of any standard notion of security**. The protocol is based on the fact that revealing the undetermined system of linear equations $\breve{A}x = y$ where $\breve{A}$ and $y$ are public does not compromise the privacy of $x$. From a security point of view, letting the adversary gain the knowledge of $\breve{A}x$ is **completely unsafe**. A clear example is already given the detailed comments subsection below for certain choices of $\breve{A}$.

Even if the paper proposes some defenses to avoid the most obvious threats in the choice of $\breve{A}$ (i.e. if the system of equations already directly exposes some coordinates of $x$), these defenses are only a minor improvement in the overall security. The protocol is in fact insecure for any $\breve{A}$. For example, consider that a party joins the aggregation protocol with a vector $x = (x_1, 0, \dots, 0)$ (i.e., a vector where $x_1$ is the only non-zero value). In this case, values of $\breve{A}x$ will always be multiples of $x_1$. Therefore, the claim of Theorem 1 does not hold: multiples of $x_1$ are *distinguishable* from uniformly random numbers, contrary to what is claimed in hybrid 1 ($H_1$) in the proof of Theorem 1 (Appendix D.1).  This renders the proof of Theorem 1 incorrect.

The computational improvements of this protocol come from the insecure modification described above. This makes the protocol inapplicable. Moreover, the attempts to further "complicate" the linear equations by the presented enhancements also follow an unsafe methodology lacking proper proofs (see my detailed comments below).

In addition to the above, the work ignores important lines of work in compression under privacy constraints (e.g. see [R1-R5] below) and differentially private-based aggregation (e.g., by the use of correlated noise [R6-R8]), directly related to the current contribution.


# Detailed Comments

- Page 2, Section 2:
    - "Mask-based" approaches are an instantiation of "SMPC-based" approaches.
    - "(i) improving the masking mechanism":  it is not clear what this means
    - "Note that the security of FL remains an open issue": this is too broad and it is not clear what "security" means in this context
- Page 3:
    -  Section 2: "However, their ability to prevent poisoning attacks is limited (Ma et al., 2023)": not sure how the reference is relevant here. Does  (Ma et al., 2023) provides evidence about this statement?
    - Section 3,  "ultimately imposing significant computational burdens": "burdens" $\rightarrow$ "burden"; is this computational burden significant with respect to the computational cost of local training steps required by ML?
- Page 4:
    - Def 1: "... where $AK$ denotes the additional knowledge ..": so far no mention of "additional knowledge has been made", so it is not clear to what this refers. Also, it should be explicitly clarified that $rank(A, Ax)$ means the rank of the horizontal concatenation of $A$ and $Ax$.
   - "... rendering it impossible to determine that specific confidential vector." this is an overly strong statement (at least if no additional context is given). Consider for example that $A$ equals the identity matrix. Indeed $\breve{A}x$ has infinite solutions (all possible values of the removed coordinate of $x$). However, almost all coordinates of $x$ will be revealed if $\breve{A}x$ is revealed.
- Page 5, Sec 3.3:
    -  "... even if some secagg entries are compromised": There is no motivation of the extra defense, explaining how these entries would be compromised.
    - "complicating the relationships among entries and further enhancing privacy": This lacks a proof. Providing privacy by obscurity (i.e, providing an obfuscation technique without proving that indeed it reduces to hard problem for the adversary) is a bad practice in the field of security.


# References

[R1] Bassily, Raef, and Adam Smith. “Local, Private, Efficient Protocols for Succinct Histograms.” In Proceedings of the Forty-Seventh Annual ACM Symposium on Theory of Computing, 127–35. STOC ’15. New York, NY, USA: Association for Computing Machinery, 2015. https://doi.org/10.1145/2746539.2746632.

[R2] Feldman, Vitaly, and Kunal Talwar. “Lossless Compression of Efficient Private Local Randomizers.” In Proceedings of the 38th International Conference on Machine Learning, 3208–19. PMLR, 2021. https://proceedings.mlr.press/v139/feldman21a.html.

[R3] Liu, Yanxiao, Wei-Ning Chen, Ayfer Özgür, and Cheuk Ting Li. “Universal Exact Compression of Differentially Private Mechanisms.” arXiv, May 28, 2024. https://doi.org/10.48550/arXiv.2405.20782.

[R4] Shah, Abhin, Wei-Ning Chen, Johannes Ballé, Peter Kairouz, and Lucas Theis. “Optimal Compression of Locally Differentially Private Mechanisms.” In Proceedings of The 25th International Conference on Artificial Intelligence and Statistics, 7680–7723. PMLR, 2022. https://proceedings.mlr.press/v151/shah22b.html.

[R5] Triastcyn, Aleksei, Matthias Reisser, and Christos Louizos. “DP-REC: Private & Communication-Efficient Federated Learning.” arXiv, December 7, 2021. https://doi.org/10.48550/arXiv.2111.05454.

[R6] Imtiaz, Hafiz, Jafar Mohammadi, and Anand D. Sarwate. “Distributed Differentially Private Computation of Functions with Correlated Noise.” arXiv, February 22, 2021. https://doi.org/10.48550/arXiv.1904.10059.

[R7] Kairouz, Peter, Brendan Mcmahan, Shuang Song, Om Thakkar, Abhradeep Thakurta, and Zheng Xu. “Practical and Private (Deep) Learning Without Sampling or Shuffling.” In Proceedings of the 38th International Conference on Machine Learning, 5213–25. PMLR, 2021. https://proceedings.mlr.press/v139/kairouz21b.html.

[R8] Sabater, César, Aurélien Bellet, and Jan Ramon. “An Accurate, Scalable and Verifiable Protocol for Federated Differentially Private Averaging.” Machine Learning 111, no. 11 (November 1, 2022): 4249–93. https://doi.org/10.1007/s10994-022-06267-9.

**Questions:**

Could please you address the points raised in "Main Weaknesses" above?

In addition to these questions:
- Could you illustrate in more detail what which masking operations of the compared SecAgg protocols do you avoid by the use of your proposal? It seems that either performing a matrix-vector multiplication or masking operations does not eliminate the dependency of the dimension of a vector $x$ in the computation.

- If we compare the computational cost of the SecAgg protocol and the computational cost of a local training step for a client, what proportion of the computation the SecAgg overhead represents? How does this change for different models?

---

> ### Author Response · Authors · 2024-11-19
>
> ## Response to the Issues of Privacy
> The statement "*In this case, values of $\check{\mathbf{A}}\mathbf{x}$ will always be multiples of $x_1$*" has caused me considerable confusion. **How can a vector be a multiple of a scalar?** Using the parameters from General Response 1 as an example, suppose the user vector is $\mathbf{x}=(7,0,0)$ and $\mathbf{y}=\check{\mathbf{A}}\mathbf{x}=(42,56)$. Could you please provide further clarification using this example? Our approach differs from the compression-based method you mentioned. **Their focus is orthogonal to ours**, and we have already demonstrated in our paper that our compression is lossless (if without DVE).
> ## Response to Detailed Comments
> 1. The mask-related approaches indeed belong to SMPC. However, due to the broad implications of mask-related approaches, many works [1][2] classify mask-based solutions separately. "*(i) improving the masking mechanism*" refers to the introduction of novel masking mechanisms as proposed in [2][3]. "*The security of FL*" pertains to safeguarding the privacy of user inputs.
> 2. Please refer to **Input correctness** in [4] on page 3: "*...providing strong guarantees against malicious inputs remains an open problem. Some works use zero-knowledge proofs to bound how much a client can bias the final result, but they are unable to formally prove the absence of all possible attacks.*" Similarly, establishing strong constraints against malicious inputs remains an unresolved challenge, and **it falls beyond the scope of our work**. For the end-to-end comparison, please refer to Fig. 2. The local training steps required by ML are independent of the overhead of secure aggregation, and the overhead of the local training steps is **not the focus of this paper**.
> 3. $AK$ is addressed in Lemma 1 and Lemma 2 of Appendix D.2 (Eq. 21 and Eq. 22). $\mathbf{A}$ undergoes certain pre-checks (see Sec. 4.1). Since $\mathbf{A}$ is public and agreed upon by all clients,  any malicious construction would require collusion among all users and can be detected by the honest users. Clearly, an overly simplistic $\mathbf{A}$ cannot be used.
> 4. "... even if some secagg entries are compromised": We are considering extreme cases here. "complicating the relationships among entries and further enhancing privacy": We have not employed obfuscation techniques. Our objective is to ensure that $\mathbf{y}_j$ is not solely related to the privacy vector $\mathbf{x}_j$ of the current $j$-th group, but also to elements from other groups (as shown in Eq. 8). This is to prevent the leakage of certain elements within the current group from undermining the security of other elements.
> 5. Note that PVF does not attempt to alter SAP or eliminate the dependency of the dimension of a vector, it reduces the number of entries processed in SAPs while ensuring intact aggregation of all entries in the original vector. Furthermore, the experimental results of PVF are highly significant.
> 6. The proportion of SecAgg overhead in the computational cost of a local training step for a client is **unrelated** to the goal of reducing SAP computational overhead in this paper.
> ## Reference
> [1] Liu, Ziyao, et al. "Privacy-preserving aggregation in federated learning: A survey." IEEE Transactions on Big Data (2022).
>
> [2] Liu, Ziyao, et al. "Efficient dropout-resilient aggregation for privacy-preserving machine learning." IEEE Transactions on Information Forensics and Security 18 (2022): 1839-1854.
>
> [3] Stevens, Timothy, et al. "Efficient differentially private secure aggregation for federated learning via hardness of learning with errors." 31st USENIX Security Symposium (USENIX Security 22). 2022.
>
> [4] Ma, Yiping, et al. "Flamingo: Multi-round single-server secure aggregation with applications to private federated learning." 2023 IEEE Symposium on Security and Privacy (SP). IEEE, 2023.

---

> ### Comment · Reviewer_r5Dg · 2024-11-19
> **Security Problems**
>
> Let me answer your question regarding the major concerns (i.e., the insecurity of the protocol).
>
> You ask "**How can a vector be a multiple of a scalar?**": the answer to your question is right there in the example you provide. It is clear that all the elementss of vector $\mathbf{y}$ are multiple of $7$ (i.e., $42= 7\times 6$, $56=7\times 8$). In fact this will happen for any $\check{A}$ you choose. Therefore, the distribution of $\mathbf{y}$ will assign probability equal to $0$ to all numbers that are not multiples of $7$. This distribution is then **clearly** distinguishable from uniformly random numbers, making (as said in my review) your claim for hybrid 1 (and therefore Theorem 1) incorrect.
>
> This is one way to easily illustrate the security problems, showing the strong dependence between the private data and $\mathbf{y}$. Other broader arguments that align with mine are the ones provided by Reviewers zANE and SAYu.

---

> > ### Author Response · Authors · 2024-11-19
> >
> > Given the private vector $\mathbf{x} = (x_1, x_2, x_3) = (7, 0, 0)$, it is indeed evident that all elements of the vector $\mathbf{y}$ are multiples of 7. However, from the server's perspective (the server obviously does not know $\mathbf{x}$), it only has access to $\mathbf{y}$, which is expressed as:
> > $$
> > \begin{array}{lll}
> > 6x_1+9x_2+5x_3=42, \\\\
> > 8x_1+4x_2+x_3=56,
> > \end{array}
> > $$
> > For instance, the server cannot distinguish whether $\mathbf{x}$ is $(7, 0, 0)$ or $(18, -34, 48)$. Additionally, we have revised H1, please refer to the latest version of the PDF.

---

> > > ### Comment · Reviewer_r5Dg · 2024-11-20
> > >
> > > Thanks for your response.
> > >
> > >  - Just to understand a bit more the revised version, could you clearly explain what did you modify in $H_1$ and why did you make that change?
> > >
> > > I understand your comment. However, proving security requires to prove something much stronger than just informally showing that the adversary cannot immediately get the private values as you argue.
> > >
> > > Following standard methodology on security (e.g., such as sequentially composable security [R1] or universal composability [R2]), for a multiparty protocol (in this case, secure aggregation) to be secure, it is required that the view of the execution does not reveal more information than the ideal functionality (in this case a functionality that only reveals the sum $\sum_{i \in \mathcal{U} \setminus \mathcal{C}} {\mathbf{x}^i}$ of the private vectors of the honest parties).
> > >
> > > This allows that protocols obtain *well studied* properties such as not having unexpected leakages when executed multiple times as in FL frameworks or in parallel as in higher level multiparty computations. This is why it is required that a simulator in the ideal world can emulate the protocol in the real world **without other information about private values other than its sum (i.e., the output of the ideal functionality)**.
> > >
> > > On the other hand, the protocol you propose reveals by construction the majority of the linear relations between the elements of private vectors. This narrows the space of search of the adversary exponentially in the dimension of the private vectors, plus the information about  $\sum_{i \in \mathcal{U} \setminus \mathcal{C}} {\mathbf{x}^i}$.  This is overwhelmingly more information than aggregation protocols that are proven secure under standard notions (e.g., [R3, R4]). Given the amount of information that your protocol reveals, it seems to me that it is indeed insecure and therefore it not fair to be compare it with other secure protocols.
> > >
> > > While the above problems already hold for semi-honest adversaries that follow the protocol, you claim to prove security for a malicious adversary. However, your proof in Theorem 1 does not contemplate common steps required for such a proof (e.g., what happens in the simulation if the adversary sends corrupted messages).
> > >
> > > - Could you specify more clearly what does Theorem 1 ensures and in which security framework the proof relies?
> > >
> > > [R1] Oded Goldreich. 2009. Foundations of cryptography: volume 2, basic applications.
> > > Cambridge university press, Cambridge, England.
> > >
> > > [R2] Canetti, Ran. "Universally composable security: A new paradigm for cryptographic protocols." Proceedings 42nd IEEE Symposium on Foundations of Computer Science. IEEE, 2001.
> > >
> > > [R3] Bell, James Henry, et al. "Secure single-server aggregation with (poly) logarithmic overhead." Proceedings of the 2020 ACM SIGSAC Conference on Computer and Communications Security. 2020.
> > >
> > > [R4] Bonawitz, Keith, et al. "Practical secure aggregation for privacy-preserving machine learning." proceedings of the 2017 ACM SIGSAC Conference on Computer and Communications Security. 2017.

---

> > > > ### Author Response · Authors · 2024-11-21
> > > > **Refined Proof of Theorem 1**
> > > >
> > > > Thank you for your detailed response and valuable comments.
> > > >
> > > > In our previous revision of the PDF, we inadvertently omitted the content related to DVE in the proof of Theorem 1 (old H_1) for clarity. This was an oversight, and you may refer to the latest version of the document to review the complete revised proof of Theorem 1 in Appendix D.3 (p17). Here, we provide a brief summary of the changes.
> > > >
> > > > The security framework we rely on is the Universal Composability (UC) framework.
> > > >
> > > > * Firstly, in Lemma 3, we provide theoretical support indicating that PVF does not leak private elements or their relationships, based on the hardness of the Learning With Errors (LWE) decision problem. And we have refined the proof of Theorem 1 ($H_3$).
> > > >
> > > > * Secondly, we have included a security analysis under the active adversary model. In Appendix D.2, we have added an introduction to cryptographic primitives, specifically symmetric authenticated encryption and digital signatures. And we have refined Figure 13 and the proof of Theorem 1 ($H_1$-$H_2$). It is evident that, in the active adversary model, the adversary's attempts to forge messages from other honest participants or to maliciously construct messages will not result in the leakage of user privacy. And the user privacy in the SAP protocol is ensured by the integrated SAP itself.
> > > >
> > > > If you have any further questions, please do not hesitate to raise them.

---

> > > > > ### Comment · Reviewer_r5Dg · 2024-11-22
> > > > >
> > > > > Thank you for your answers.

---

### Official Review · Reviewer_9mLH · 2024-11-03

**Soundness:** 2
**Presentation:** 2
**Contribution:** 3
**Rating:** 6
**Confidence:** 3

**Summary:**

This paper devises a portable module named \lamba-SecAgg for secure aggregation in federated learning. The authors also propose an extension involving disrupting variables to enhance privacy. Through extensive experiments, they demonstrate the efficiency of the proposed method, achieving up to 99.5× speedup and up to 32.3× communication reduction.

**Strengths:**

1.Theoretical proofs.
2.The experimental results demonstrate that PVF achieved 99.5 \times speedup and up to 32.3 \times communication reduction.

**Weaknesses:**

1. Writing/technical issues:
(1) In the Introduction section, the author methioed that "the minimal noise added by DP is insufficient to thwart attacks", yet they also suggest considering "DP in the extension for enhanced privacy." Is there a deeper reason or gap that I might have overlooked?
(2) The introduction of "compression-based techniques" in Figure 3 and Section 2 feels somewhat abrupt, primarily due to the lack of clarity in the classification of existing solutions outlined in the Introduction section. I suggest providing a clearer explanation of the criteria used to categorize the methods into secure aggregation techniques and compression-based techniques. Additionally, providing an analysis of the limitations of existing methods would help readers better understand the motivation behind the development of the PVF.
(3) The definition of adversary in threat model is not very clear, particularly regarding key aspects such as adversary knowledge and adversary capabilities, which have not been adequately explained or defined.
(4) Figure 4 is too abstract to understand.
(5) In Section 3.3, while discussing secure aggregation, it is noted that the requirements for data accuracy are relatively high. However, the introduction of DP typically involves adding noise to the data. It would be beneficial to clarify how the accuracy of the data can be maintained after noise has been added, particularly in the context of the freezing and melting processes.

2. Experimental issues:
(1)The neural network architectures and datasets are not intruduced in the ‘Experimental settings’.
(2)The setting of (\lambda = 100) in some experiments requires further explanation.
(3)The experimental validation, although comprehensive, is limited to specific neural network architectures and datasets. Its generalisability to other models and types of data may require further examination.

**Questions:**

Please refer to the weakness above.

---

> ### Author Response · Authors · 2024-11-19
>
> We express our sincere gratitude for your recognition and support!
> 1. About the expression of DP. What we want to express is that relying **solely** on the minimal noise added by DP is insufficient to thwart attacks, and our solution does not rely solely on DP to ensure privacy.
> 2. Compression-based baselines. These baselines are the closest in nature to our proposal, so we list them separately.
> 3. Threat model. Our threat models (please refer to line 145) are
>    1.  Semi-honest Model[1], also referred to as the "honest-but-curious" model, where participants in a protocol are assumed to follow the protocol correctly but may try to extract additional information from the data they receive. In this model, adversaries do not deviate from the protocol's rules or engage in malicious behavior like collusion or input manipulation. However, they may attempt to infer sensitive data from the information they have access to, using computational resources to gain an advantage. This model is often used in cryptography and secure multiparty computation, where participants are trusted to some extent but must be safeguarded from exploiting any inadvertent information leakage.
>    2. Active Adversary Model[1], also known as the "malicious adversary" model. In this scenario, adversaries are not only capable of following the protocol but can also actively deviate from it, forging messages, manipulating inputs, or colluding with other participants to subvert the protocol. This model assumes that adversaries might engage in arbitrary, harmful behavior with the intention of compromising the security or correctness of the system.
>
> 4. General Response 1 provides a simple example, which we hope will assist you in understanding Fig.4.
> 5. Data Accuracy. We add a small amount of DP noise to the original vector before PVF. After PVF and SAP, we get the aggregation result with a little noise, which is consistent with the LDP-based SAP. Due to the existence of SAP, a small $\sigma$ is sufficient. Consistent with [2], we set the standard deviation of the noise to 0.0409 in Fig. 6.
> 6. Experimental issues. \(\lambda = 100\) refers to the parameter \(\lambda\) in PVF. In the "End-to-end comparison" in Sec. 5.3 (line 405) and "Disrupting Variables Extension" in Sec. 5.4 (line 446), we introduce the datasets and models used. In other experiments, such as those in Tab. 1 and Fig. 5, the user's original vectors are randomly generated. This is because these experiments primarily focus on the overhead of the secure aggregation phase, and the attributes of models (apart from the total parameter count) and datasets do not influence the results, as consistently pointed out in previous works [3][4].
> ## Reference
> [1] Bonawitz, Keith, et al. "Practical secure aggregation for privacy-preserving machine learning." proceedings of the 2017 ACM SIGSAC Conference on Computer and Communications Security. 2017.
>
> [2] Stevens, Timothy, et al. "Efficient differentially private secure aggregation for federated learning via hardness of learning with errors." 31st USENIX Security Symposium (USENIX Security 22). 2022.
>
> [3] Hahn, Changhee, et al. "VerSA: Verifiable Secure Aggregation for Cross-Device Federated Learning." IEEE Transactions on Dependable and Secure Computing 20.1 (2023): 36-52.
>
> [4] Liu, Ziyao, et al. "Efficient dropout-resilient aggregation for privacy-preserving machine learning." IEEE Transactions on Information Forensics and Security 18 (2022): 1839-1854.

---

### Official Review · Reviewer_a2uE · 2024-11-03

**Soundness:** 3
**Presentation:** 2
**Contribution:** 2
**Rating:** 5
**Confidence:** 2

**Summary:**

The paper addresses the challenges of secure aggregation in federated learning, particularly the high computation costs associated with Secure Aggregation Protocols (SAPs). The paper introduces a novel approach called Partial Vector Freezing (PVF), designed to reduce computation without increasing communication overhead. In addition, the paper proposes the disrupting variable extension to PVF to support enhanced privacy. The extensive experiments show the effectiveness of the proposed proposal.

**Strengths:**

1. The PVF significantly compresses the length of the vector involved in SAP.
2. The disrupting variables extension method improves privacy, without the computational overhead.
3. The authors conduct extensive experiments

**Weaknesses:**

1. Lack of Novelty in the Proposed Solution:
While I appreciate the clarity and straightforwardness presented in your methodology, I am concerned about the apparent simplicity of the proposed solution. The approach, as described, seems to lack the level of innovation. Consider expanding on the theoretical background, comparing your method with others in detail, and emphasizing any novel insights or improvements that your solution offers.
2. Informality in Security Analysis:
The security analysis section of your paper appears to be somewhat informal and lacks the rigor typically required for a comprehensive evaluation of a proposed system or method. Security is a critical aspect in many research domains, and a thorough, formal analysis is essential to establish trustworthiness and robustness. I recommend conducting a more structured and detailed security analysis, possibly incorporating formal security proofs, case studies, or simulations to demonstrate the effectiveness of your security measures.

**Questions:**

1. In practical applications, how should this value \lambda be determined?

2. Are there any fundamental differences between the aggregation method of k^i and that of y^i?

---

> ### Author Response · Authors · 2024-11-19
>
> ## Response to Questions
> 1. Please refer to the second paragraph of Appendix E.1 in our paper.
> 2. Please read Sec. 3.2, specifically **the Phase 2: Main.SecAgg(·) section**.

---

### Official Review · Reviewer_z5KJ · 2024-11-04

**Soundness:** 2
**Presentation:** 2
**Contribution:** 2
**Rating:** 5
**Confidence:** 3

**Summary:**

This paper introduces a novel method called λ-SecAgg, which integrates a module named Partial Vector Freezing (PVF) into Secure Aggregation Protocols (SAPs) for federated learning. The main goal of this method is to reduce the computational overhead by “freezing” most of the entries in user update vectors, allowing only a fraction (1/λ) of the original vector to be processed through secure aggregation. The frozen entries can later be “thawed” to recover the full aggregated vector, ensuring that no information is lost in the final aggregation. Additionally, the paper proposes a Disrupting Variables Extension (DVE) that enhances privacy by adding noise to the frozen entries using Differential Privacy (DP). The authors perform extensive empirical evaluations across seven baselines, demonstrating that PVF can achieve up to 99.5× speedup and 32.3× communication reduction without compromising user privacy or security.

**Strengths:**

Innovation: The concept of freezing and unfreezing vector entries in the context of secure aggregation is very novel. This approach effectively reduces the computational burden on SAP, which has been a significant bottleneck in real-world federated learning applications, especially for large-scale models such as Large Language Models (LLMs).

Comprehensive Evaluation: The authors evaluate their approach on seven different baselines covering various secure aggregation protocols (e.g., homomorphic encryption-based, SMPC-based, mask-based). The experimental results show substantial improvements in computation time and communication cost.

Privacy and Security: The paper proves the privacy guarantees of λ-SecAgg under semi-honest and active adversary models through security analyses. In addition, the authors introduce extensions such as DVE, which further strengthens the privacy guarantees.

**Weaknesses:**

Clarity and readability: Although this paper presents a novel approach, some sections are dense and difficult to understand, especially the mathematical derivations and safety analyses. It is suggested that the authors could improve the readability of these sections by providing more intuitive explanations and breaking down the steps as much as possible. In addition, the readability of some diagrams and formulas (e.g., those in Sections 3 and 4) is too low, and it is suggested that the reader could improve them by simplifying them or providing more detailed explanations.

Impact of noise on accuracy: Although the paper claims that the impact of DVE (adding noise to DP) on accuracy is negligible, the experimental results on the loss of accuracy due to DP noise are not detailed enough. It is suggested that the authors can add relevant experiments for this part.

Scalability to Multiple Users: This paper focuses on performance improvements for single users and servers, but does not discuss scalability to multiple users. It is suggested that the authors validate the approach of this paper in the context of multiple simultaneous users, especially with respect to communication overheads and system latency.

**Questions:**

This paper presents a novel and practical approach to reduce the computational overhead of secure aggregation in federated learning by proposing λ-SecAgg with partial vector freezing (PVF). The strengths of the method lie in its innovative design, theoretical rigor and comprehensive evaluation, showing significant performance improvements. However, there are areas that could benefit from further clarification, particularly in terms of readability, real-world evaluation, and the impact of noise on accuracy. It is recommended that the authors consider the above comments to further refine and optimize the paper.

---

> ### Author Response · Authors · 2024-11-19
>
> Thank you for your recognition and hope the response can address your concerns.
>
> 1. Clarity and readability. General Response 1 provides a simple example, which we hope will assist you in better understanding PVF.
> 2. Impact of noise on accuracy. Note that Differential Privacy (DP) is an extension of PVF. The experiments on DP are not the primary focus of this paper. As demonstrated in the experiments in [1][2], the experiments shown in Fig. 6, conducted in two scenarios, are sufficient to demonstrate that the impact of DP noise on accuracy is negligible.
> 3. Scalability to Multiple Users. All experiments in our paper focus on exploring multi-user scenarios. Please refer to the setup of each experiment (typically indicated in the titles of figures and tables).
> ## Reference
> [1] Stevens, Timothy, et al. "Efficient differentially private secure aggregation for federated learning via hardness of learning with errors." 31st USENIX Security Symposium (USENIX Security 22). 2022.
>
> [2] Liu, Zizhen, et al. "SASH: Efficient secure aggregation based on SHPRG for federated learning." Uncertainty in Artificial Intelligence. PMLR, 2022.

---

### Official Review · Reviewer_zANE · 2024-11-04

**Soundness:** 2
**Presentation:** 3
**Contribution:** 2
**Rating:** 3
**Confidence:** 4

**Summary:**

The authors present a new system to improve the computational overhead of secure aggregation using a new approach called Partial Vector Freezing. This approach reduces the number of entries processed in the secure aggregation protocol, by projecting chunks of the client input vector onto a different space, and only aggregating $1/\lambda$ of the entries of each chunk securely and sending the rest of the entries in the clear. The server aggregates the entries from all clients and recovers the original input vectors by projecting the inputs back to the original space. The paper further bolsters privacy through the Disrupting Variables Extension, which applies noise calibrated for Local Differential Privacy to frozen vectors. Experimental results demonstrate substantial computation improvements compared to state-of-the-art secure aggregation protocols.

**Strengths:**

- The focus of privacy-preserving federated learning is a crucial topic
- Extensive evaluation that covers a wide range of existing secure aggregation protocols

**Weaknesses:**

- The approach impacts the robust privacy guarantees traditionally upheld by state-of-the-art secure aggregation protocols. These protocols typically ensure that an adversary gains no additional information about the inputs of honest clients beyond what is inferred from the aggregated output. However, Partial Vector Freezing (PVF) significantly reduces this privacy. As pointed out by the authors, it is possible for the server to learn whether two clients have similar vector chunks. Although the authors propose a mitigation strategy through Local Differential Privacy to reduce the detection of exact matches, this measure does not fully mitigate the issue of input privacy. The noised client inputs may still leak partial information that allows the server to deduce similarities between inputs. Given this trade-off, the computational gains provided by PVF do not justify the notable privacy impact. For instance, in the context of PracAgg, the masking computation is relatively lightweight. It involves field operations and pseudorandom generator evaluations, typically implemented with efficient cryptographic functions like AES. Additionally, the more computationally intensive pairwise key agreements are independent of the vector size and remain necessary regardless of the implementation of PVF.
- Another concern is the soundness of the security proof presented in Theorem 1. Specifically, the claim that the protocol execution is indistinguishable from random simulation seems to be inaccurate. The distribution of Hybrid 1 is not indistinguishable from that of Hybrid 0, as the distribution of frozen vectors $y_i$ does not exhibit properties of uniformly sampled vectors. While the random vectors are sampled uniformly from $\mathbb{Z}_p$, the frozen vectors in the protocol are the actual inputs masked with centered Gaussian noise of bounded variance. This results in a non-uniform distribution over $\mathbb{Z}_p$ undermining the indistinguishability between the two hybrids. Furthermore, other parts of the security proof are incomplete.  For instance, in Hybrid 3, it is stated that the adversary-controlled clients $\mathcal{C}$ call the ideal functionality. However, in simulation-based proofs, it is typically the simulator, not the adversary, that has direct access to the ideal functionality. Clarifying this aspect would strengthen the proof’s rigor and ensure alignment with standard cryptographic practices.

**Questions:**

1. The baseline runtime figures for the secure aggregation protocols presented in Figure 1 and Table 1 appear notably higher than those reported in related literature. For instance, in the case of PracAgg with a vector length of 100k elements, Figure 1 shows a client runtime of 14 seconds and a server runtime of 140 seconds. In contrast, the original paper by Bonawitz et al. (2017) reports significantly lower runtimes for similar conditions, with client runtimes around 300 milliseconds (Figure 6a) and server runtimes at most 5 seconds (Figure 7a). Could you clarify the reasons for this discrepancy in runtime comparisons?
2. Could you provide a more detailed analysis of the privacy impact of your scheme, particularly focusing on the amount of differentially private noise that would be sufficient to mitigate privacy risks effectively? A clearer discussion on how the noise level was determined and its implications on both privacy and utility would be valuable.

---

> ### Author Response · Authors · 2024-11-19
>
> We would like to appreciate you for your constructive feedback. We address your questions and concerns in the following.
>
> 1. Privacy issues of two identical input. After introducing DVE, Eq. 8 ensures that when the noise of $\mathcal{N}(0,\sigma^2)$ is only added to $\mathbf{x}$, the noise of $\mathcal{N}(0,(1+\frac{l}{\lambda})\sigma^2)$ is added to $\mathbf{y}$. For example:
>    $$A_{1,1} (x_{(j-1) \lambda + 1} + \underline{k_1 +\cdots+ k_{\left \lfloor \frac{l}{\lambda} \right \rfloor }})+\cdots+A_{1, \lambda} (x_{j\lambda} + \underline{k_{l- \left \lfloor \frac{l}{\lambda} \right \rfloor} + \cdots + k_l})   =y_{(j-1) \lambda + 1}.$$
>    The additional noise does not affect the recovery of $\mathbf{x}$, as it is eliminated during the **thawing** process, i.e.,
>    $$A_{1,1} (\sum x_{(j-1) \lambda + 1} )+\cdots+A_{1, \lambda} (\sum x_{j\lambda} )=\sum y_{(j-1) \lambda + 1} - \underline{ \sum \sum_{o\in [1, \lambda]} A_{1,o}\sum_{r\in \left[(o-1)\left \lfloor \frac{l}{\lambda} \right \rfloor+1,o\left \lfloor \frac{l}{\lambda} \right \rfloor \right]} k_{r} }.$$
>    Moreover, as pointed out in PracAgg[1] on page 12 of their paper: "*...almost all of the computation cost comes from expanding the various PRG seeds to mask the data vector. Compared to this, the computational costs of key agreement, secret sharing and reconstruction, and encrypting and decrypting messages between clients, are essentially negligible.*" Therefore, the efficiency gains brought by PVF are of considerable importance.
> 2. The indistinguishability between $H_0$ and $H_1$. Here, we no longer prove the randomness of $y^i$. we modify "*we replace the frozen vectors $\{\mathbf{y}^{i}\} _ {i\in\mathcal{U}}$ received by $\mathcal{S}$ with uniformly random vectors.*" in H1 of Theorem 1 to "*replace $\{\mathbf{x}^{i}\} _ {i\in\mathcal{U}}$ with random vectors that maintain the same correlation between $\{\mathbf{x}^{i}\} _ {i\in\mathcal{U}}$ and $\{\mathbf{y}^{i}\} _ {i\in\mathcal{U}}$*". This adjustment ensures the continued validity of Theorem 1. Please see the modification of the proof Theorem 1 in the document. We proceed to demonstrate that the original vector remains random even with the knowledge of the correlation. Let $\mathbf{x}$ denote a fragment of an original vector. We denote the **general solution** obtained by the adversary as $\mathbf{x'}=\rho\mathbf{a}+\mathbf{b}$, where $\mathbf{a}$ and $\mathbf{b}$ are solved given $\mathbf{y}^i$, $\rho$ is an uncertain variable. The correct $\rho$ corresponds to $\mathbf{x}$ is $ \rho _ * $, i.e., $\mathbf{x} = \rho _ * \mathbf{a}+\mathbf{b}$. It can be seen that $\mathbf{x'}$ differs from $\mathbf{x}$ in both magnitude and direction. That means upon acquiring the correlation, the server essentially gains that $\mathbf{x}$ is still **randomly distributed with its initial point and terminal point on two parallel lines respectively** (as visually apparent from Fig. 9 in the new PDF), which does not compromise privacy.
> 3. The call to the ideal function. We expressed it correctly in the main text (line 309). Thank you for pointing out the typo in the appendix.
> 4. **Please note that the server's computation time in PracAgg is not 5 seconds, but rather 50 seconds**. Authors of PracAgg do not provide the source code, but we have strictly followed the implementation outlined in their proposal, as detailed in Appendix E.2 in our paper. We speculate that this discrepancy might be due to differences in machine configurations. The results in Ref[2][3] are similar to those in our paper, with the client computation time being around 10 seconds and the server overhead approximately 100 seconds.
>
> ## Reference
> [1] Bonawitz, Keith, et al. "Practical secure aggregation for privacy-preserving machine learning." proceedings of the 2017 ACM SIGSAC Conference on Computer and Communications Security. 2017.
>
> [2] Hahn, Changhee, et al. "VerSA: Verifiable Secure Aggregation for Cross-Device Federated Learning." IEEE Transactions on Dependable and Secure Computing 20.1 (2023): 36-52.
>
> [3] Liu, Ziyao, et al. "Efficient dropout-resilient aggregation for privacy-preserving machine learning." IEEE Transactions on Information Forensics and Security 18 (2022): 1839-1854.

---

> > ### Author Response · Authors · 2024-11-23
> >
> > Dear Reviewer zANE:
> >
> > In Lemma 3, we offer theoretical support demonstrating that PVF does not leak private elements or their relationships, based on the hardness of the Learning With Errors (LWE) decision problem. And we have refined the proof of Theorem 1 ($H_3$). We hope this addresses your concerns regarding the privacy. A concise summary of the progress made in the rebuttal can be found in General Response 2. If you have any further questions, please do not hesitate to contact us.
> >
> > We look forward to receiving your updates.

---

> > > ### Comment · Reviewer_zANE · 2024-11-25
> > >
> > > Dear Authors,
> > >
> > > Thank you for providing clarifications. After carefully reviewing your responses regarding the privacy guarantees, as well as the updated manuscript, I remain unconvinced about the robustness of your protocol's privacy guarantees and the soundness of the presented security proof.
> > >
> > > For example, while Theorem 1 still claims to address malicious adversaries, the proof does not adequately handle these scenarios. Similarly, replacing all plaintext inputs with dummy messages in H1 likely changes the output distribution compared to H0. As a result, the paper fails to demonstrate a satisfactory trade-off between performance and privacy in its current state, and significant changes are required to ensure the soundness of the proof.

---

> ### Author Response · Authors · 2024-11-25
>
> Dear Reviewer zANE:
>
> Thank you for your comment. We hope the following response can help clarify your concerns.
>
> ### **1. Privacy in the Active Adversary Model**
> In the active adversary model, malicious participants may:
>
> * send malformed or incorrect messages to disrupt the calculations of honest parties.
> * forge fake users to engage in the protocol.
> * attempt to forge or tamper with the messages of other parties.
> * attempt to send a fabricated special message.
>
> All of these attacks can be avoided by the use of symmetric authenticated encryption and digital signatures (as in $H_1$ and $H_2$ of our proof). This security assurance is also evident in the security analysis of other aggregation protocols, such as the proof of Theorem IV.4 in [1] and the proof of Theorem A.2 in [2].
>
> ### **2. Indistinguishability between $H_1$ and $H_0$**
> In fact, replacing all plaintext inputs with dummy messages in $H_1$ **does not** change the distribution compared to $H_0$. This is because the symmetric authenticated encryption we employ satisfies both IND-CPA (Indistinguishability under Chosen Plaintext Attack) and INT-CTXT (Integrity of Ciphertexts), as stated in **line 878** of our paper. Let's briefly introduce these two security requirements:
>
> * IND-CPA ensures that an encryption scheme maintains indistinguishability under chosen plaintext attacks[3]. It means that even if an attacker can select arbitrary plaintexts and observe their corresponding ciphertexts, they cannot deduce any information related to the plaintext from the ciphertext. In short, the attacker is unable to distinguish between ciphertexts with any significant advantage over random guessing, regardless of the plaintext encrypted.
>
> * INT-CTXT guarantees that an attacker, without knowledge of the plaintext or the key, cannot generate a valid ciphertext that, when decrypted, results in a legitimate plaintext[3].
>
> Thus, after replacing the plaintext in $H_0$ with dummy messages, SIM is unable to distinguish the ciphertexts after encryption. This point is widely acknowledged in the security aggregation literature, as evidenced by the proof of Theorem 6.3 in [2], specifically $Hyb_2$, as well as $Hyb_4$ and $Hyb_5$ of the proof of Theorem A.2 in [2].
>
> If you have any other questions, please let us know.
>
> ### Reference
> [1] Liu, Ziyao, et al. "Efficient dropout-resilient aggregation for privacy-preserving machine learning." IEEE Transactions on Information Forensics and Security 18 (2022): 1839-1854.
>
> [2] Bonawitz, Keith, et al. "Practical secure aggregation for privacy-preserving machine learning." proceedings of the 2017 ACM SIGSAC Conference on Computer and Communications Security. 2017.
>
> [3]Bellare, Mihir, and Chanathip Namprempre. "Authenticated encryption: Relations among notions and analysis of the generic composition paradigm." International Conference on the Theory and Application of Cryptology and Information Security. Berlin, Heidelberg: Springer Berlin Heidelberg, 2000.

---

> ### Comment · Reviewer_zANE · 2024-11-25
>
> 1. Your security proof does not account for actively malicious behavior. I am not the first reviewer to point this out (e.g., see https://openreview.net/forum?id=E1Tr7wTlIt&noteId=iog03MH6Wy). See for instance the Theorem A.2 in your reference [2] for additional steps required to achieve malicious security in secure aggregation protocols.
> 2. In my understanding of your definition of H1, you “replace the plaintext with dummy
> messages”. This will lead to a different output of the protocol, because it does not depend on the real inputs anymore. In the hybrids you reference from [2], the replacement is done in such a way that this does not lead to a different output (e.g., only encrypted messages between parties controlled by the simulator).

---

> > ### Author Response · Authors · 2024-11-25
> >
> > Dear Reviewer zANE:
> >
> > ### **1. Privacy in the Active Adversary Model**
> > As in our response to this comment (https://openreview.net/forum?id=E1Tr7wTlIt&noteId=iog03MH6Wy), symmetric authenticated encryption and digital signatures are sufficient to ensure that, in the event of malicious participants attempting the malicious behaviors mentioned in our previous response **during PVF**, the protocol will be terminated and will not lead to privacy leakage. This security assurance is also evident in the security analysis of other SAPs, such as $H_2,H_4,H_5,H_6,H_8$ of the proof of Theorem IV.4 in [1]. And this point was not contested by Reviewer r5Dg.
> >
> > While additional steps of SAP are indeed necessary to achieve security in the active adversary model (such as *ConsistencyCheck*), **note that PVF does not alter SAP or add additional communication rounds, and it is decoupled from the specific SAP**. Consequently, the security of the remaining steps except PVF is inherently ensured by the integrated SAP itself.
> >
> > ### **2. Modification of $H_1$**
> > Thank you for highlighting the lack of clarity in our writing. In PVF, **the only plaintext involved is $\mathbf{y}^i$** (the simulation of other plaintexts in SAP is handled by the integrated SAP). Accordingly, we have revised $H_1$ as follows (we have updated in the latest version):
> >
> > *$H_1$: This hybrid is distributed similarly to the previous one, except for the following modifications. $SIM$ obtains $\sum_{i \in \mathcal{U}' \backslash \mathcal{C}}\mathbf{x}^{i}$ by calling ${Ideal} _ {{\{ \mathbf{x}^{i}\}} _ {i \in \mathcal{U}\backslash \mathcal{C}}}( {\mathcal{U}' \backslash \mathcal{C}} )$. $SIM$ aborts if there is an illegal request. We replace the ciphertexts of $\{\mathbf{y}^{i}\} _ {i\in\mathcal{U}}$ with the ciphertexts of uniformly random vectors $\{\mathbf{w}^{i}\} _ {i\in\mathcal{U}}$ that satisfy $\sum_{i \in \mathcal{U}' \backslash \mathcal{C}}\mathbf{w}^{i} = \sum_{i \in \mathcal{U}' \backslash \mathcal{C}}\mathbf{y}^{i}$. $\sum_{i \in \mathcal{U}' \backslash \mathcal{C}}\mathbf{y}^{i}$ can be can be computed from Eq. 9 based on $\sum_{i \in \mathcal{U}' \backslash \mathcal{C}}\mathbf{x}^{i}$. The IND-CPA and IND-CTXT security of symmetric authenticated encryption guarantees the distribution of this hybrid is indistinguishable from the previous one.*
> >
> > The output of SIM is identical to that of REAL, and no additional private information about the honest users is disclosed.
> >
> > If you have any other questions, please let us know.
> >
> > ### Reference
> > [1] Liu, Ziyao, et al. "Efficient dropout-resilient aggregation for privacy-preserving machine learning." IEEE Transactions on Information Forensics and Security 18 (2022): 1839-1854.

---

> ### Comment · Reviewer_zANE · 2024-11-26
>
> To be clear, this answer does not address my concerns about security of your protocol.
>
> Your proof still does not account for actively malicious adversaries. While the primitives you reference (e.g., symmetric authenticated encryption and digital signatures) are known to enhance security, your proof does not explicitly demonstrate how these are applied within your protocol to address malicious behavior. I recommend referring to theory on simulation-based proofs or established approaches in related work to illustrate how to formally account for such adversaries.
>
> Similarly, your statement that "the security of the remaining steps [...] is inherently ensured by the integrated SAP itself" is not substantiated in your security proof. Typically, such a claim would require invoking the simulator or ideal functionality of the integrated SAP explicitly. Without this, the connection between the SAP and your protocol's security remains unclear.

---

> ### Author Response · Authors · 2024-11-26
> **Response part I**
>
> Dear Reviewer zANE:
> ### **R1. The Relationship between PVF and SAP**
> From your comments, we guess you may not have fully comprehended the calculation process of PVF. Therefore, we will first provide you with a detailed explanation of the relationship between PVF and the integrated Secure Aggregation Protocol (SAP).
>
> For example, given $\mathbf{x}^i = (x_1, x_2, x_3)$, the user computes $\mathbf{y}^i = \check{\mathbf{A}}(\mathbf{x}^i + \mathbf{e}) = (y_1, y_2)$. The user only inputs $k^i =\alpha(x_3 + e_3)$ into SAP while simultaneously transmitting $\mathbf{y}^i$ as a piggyback message to the server during the protocol. Due to the hardness of LWE search and decision problem, $\mathbf{y}^i$ does not reveal any private information about $\mathbf{x}^i$.
>
> Upon completing the SAP, the user obtains $\sum \mathbf{y}^i$ and $\sum k^i$. By leveraging Eq. (9), $\sum \mathbf{x}$ can be reconstructed. And we refer to the SAP integrated with PVF as $\lambda$-SecAgg. Clearly, throughout this process, PVF does not modify the operations of SAP and remains decoupled from it. As long as the SAP can securely aggregate $\sum k^i$, it can be seamlessly integrated with PVF.
>
> ### **R2. How Cryptographic Primitives are Applied in PVF**
> The utilization of these cryptographic primitives is illustrated in Fig. 13. Here, based on the description of PVF in R1, we provide a detailed explanation of the active attacks that malicious participants can launch **within PVF** and how PVF leverages these cryptographic primitives to defend against them.
>
> First and foremost, it is evident that **in PVF**, the user **only transmits** $\mathbf{y}^i$ to the server, and the server **only sends** $\sum \mathbf{y}^i$ back to the users after SAP ends. Notably, $k^i$ and $\sum k^i$ are transmitted through the SAP, independent of PVF.
>
> 1) **Forging Fake Users to Participate in PVF.** This type of attack, also known as a *Sybil Attack*, involves fake users reporting received information to the server. Such attacks primarily target scenarios where users share secret keys among themselves but keep the keys secret from the server, like PPDL [1]. Alternatively, an attacker may attempt to forge a large number of fake users (more than $\frac{1}{3}|\mathcal{U}|$) to reconstruct users’ private keys in the secret-sharing scheme. Since PVF does not involve information that is kept secret from the server but shared among all users, and consistent with the assumption in [2] that the number of malicious users does not exceed $\frac{1}{3}|\mathcal{U}|$ (line 147), PVF is resistant to this type of attack.
>
> 2) **Attempting to Forge or Tamper with Honest Users' Messages.** Such attacks may occur in PVF in the following situations: malicious participants forging or tampering with an honest user's $\mathbf{y}^i$. This can be avoided by the digital signature $\sigma_{1}^{i}$ employed in PVF. Similarly, malicious participants may attempt to forge or tamper with $\sum \mathbf{y}^i$ sent by the server, which is prevented by the use of $\sigma_{3}$. These protections are reflected in $H_2$ of the proof.
>
> 3) **Sending Malformed Messages.** In PVF, such attacks include malicious users sending malformed ciphertexts of $\mathbf{y}^i$ or the malicious server sending malformed ciphertexts of $\sum \mathbf{y}^i$. Such attacks are prevented by the IND-CPA and IND-CTXT security of the symmetric authenticated encryption used in PVF. If decryption fails, the protocol is immediately terminated. These protections are reflected in $H_1$ of the proof.
>
> 4) **Intercepting and Stealing Private Information.** Malicious adversaries may intercept messages sent by honest users to extract private information. This is effectively avoided by the symmetric authenticated encryption employed in PVF. This protection is reflected in $H_1$ of the proof.
>
> The use of symmetric authenticated encryption and digital signatures to ensure privacy under the active adversary model is a relatively mature application in the field of secure aggregation, and our design follows these existing works. In the proof, under the UC framework, $H_1$ and $H_2$ formally explain the impossibility of active adversaries forging or tampering with messages **in PVF**, as well as the fact that any attack by an active adversary would lead to the termination of the protocol, thus demonstrating the security of PVF under the active adversary model.
>
> If you believe there are specific steps missing, please do not hesitate to offer your guidance.

---

> ### Author Response · Authors · 2024-11-26
> **Response part II**
>
> ### **R3. Security provided by the Integrated SAP**
> We believe that after reading R1, you will clearly understand the relationship between PVF and the integrated SAP. PVF and the integrated SAP operate independently and do not interfere with each other. They are completely decoupled.
>
> Therefore, when proving security, we have omitted the security analysis of the aggregation of $k^i$, **as the security of the aggregation process for $k^i$ is handled by the integrated SAP, not by PVF**. This is why we can assert that "*the security of the remaining steps, except PVF, is inherently ensured by the integrated SAP itself*".
>
> If you believe there are any deficiencies in our approach, please feel free to share your insights.
>
> ### Reference
> [1] Aono, Yoshinori, et al. "Privacy-preserving deep learning via additively homomorphic encryption." IEEE transactions on information forensics and security 13.5 (2017): 1333-1345.
>
> [2] Bonawitz, Keith, et al. "Practical secure aggregation for privacy-preserving machine learning." proceedings of the 2017 ACM SIGSAC Conference on Computer and Communications Security. 2017.

---

### Official Review · Reviewer_SAYu · 2024-11-04

**Soundness:** 1
**Presentation:** 1
**Contribution:** 2
**Rating:** 1
**Confidence:** 4

**Summary:**

This paper introduces $\lambda$-SecAgg, a secure aggregation protocol for federated learning (FL) designed to reduce computational and communication overhead through Partial Vector Freezing (PVF). This paper claims that by freezing and processing only a fraction of the private vector entries, the method significantly reduces the burden on the server and participating devices while ensuring all vector entries are eventually aggregated. To further enhance privacy, the paper incorporates Disrupting Variables Extension (DVE). The authors empirically demonstrate substantial performance gains in terms of speedup and communication reduction across various secure aggregation protocols.

While the paper presents an interesting method to reduce the overhead in secure aggregation, the privacy analysis in Section 4.1 is fundamentally flawed. The authors underestimate the information leakage from $y^{i}$, which compromises the claimed privacy guarantees.

**Strengths:**

1. This paper considers a timely and important problem in secure aggregation protocol to reduce computational and communication overhead.

**Weaknesses:**

1. Most importantly, the privacy analysis in Section 4.1, which claims no privacy leakage from $y^{i}$, is flawed. Although the paper asserts that no specific element of the original vector $x$ can be deduced directly from $y^{i}$, this does not mean there is no privacy leakage. In fact, $y^{i}$ reveals significant information about $x$. For example, in the case where $\lambda = 2$ and $x$ has two elements, the server can infer $x_1$ in terms of $x_2$ from $y^{i} = a_{11}x_1 + a_{12}x_2$. While $x_1$ cannot be fully determined without $x_2$, the conditional probability of guessing $x_1$ correctly is now $1/p$ instead of $1/{p^2}$. This reduction in entropy, $H(x)$, shows that $y^{i}$ contains valuable information, thus reducing privacy. The authors should revise the privacy analysis and clarify the impact of knowing $y^{i}$ on the security of the original vectors.

**Questions:**

Please see the comment in the weaknesses.

---

> ### Author Response · Authors · 2024-11-19
>
> 1. Firstly, in Theorem 1, we explicitly state that $\mathbf{\lambda > 2}$. And **you did not compute $\mathbf{y}$ using the correct method**, please refer to General Response 1 for clarification.
>
> 2. Secondly, the conditional probability of guessing an element (like $x_1$) is inherently $\frac{1}{p}$. **Where does $\frac{1}{p^2}$ come from?**

---

> ### Comment · Reviewer_SAYu · 2024-11-19
>
> Thanks for your response and the simple example in the general response 1.
>
> Let me clarify my comments with your example. I'd like to claim that $y^1_1 \in \mathbb{F}_p^2$ provides some information about $x^1_1 \in \mathbb{F}_p^3$ to the server even though the server cannot fully deduce $x^1_1$ from $y^1_1$.
>
> This is because $H(x^1_1)=3log(p)$ while $H(x^1_1 | y^1_1)=log(p)$. Intuitively, before the server receives $y^1_1$, the number of free variables was 3 (as the number elements in $x^1_1$ is 3). After the server knows the $y^1_1$, however, the number of free variables is reduced to 1.
>
> Please correct me if I miss something.

---

> > ### Author Response · Authors · 2024-11-19
> >
> > Thank you for your response and clarification.
> >
> > Please refer to Eq. 8 in Sec. 3.2. $\mathbf{x}$ and $\mathbf{k}$ are both free variables. If we consider the underlined part of Eq. 8 as a whole, the number of known equations in each group is $\lambda-1$, while the number of unknowns is $2\lambda$.

---

> > > ### Author Response · Authors · 2024-11-23
> > >
> > > Dear Reviewer SAYu:
> > >
> > > Regarding your concerns about privacy, we have demonstrated in the latest version of the PDF (Appendix D.3, page 18) that $\mathbf{y}^i$ does not reveal any information about $\mathbf{x}^i$. This conclusion is theoretically supported by the hardness of LWE decision problem (Lemma 3). Additionally, a brief summary of the progress made in the rebuttal for our submission can be found in General Response 2. If you have any further questions, please do not hesitate to raise them.
> > >
> > > We are looking forward to your response.

---

### Author Response · Authors · 2024-11-19
**General Response 1: A simple example of PVF**

Here we provide a simple example of PVF to help the reviewers gain a clearer understanding:
Assume the model vectors for two users are $\mathbf{x^1}=(1,2,3,4,5,6,7,8,9)$ and $\mathbf{x^1}=(9,8,7,6,5,4,3,2,1)$, we set $\lambda=3$, meaning we divide $\mathbf{x}$ into $\frac{9}{\lambda}=3$ groups. Following PVF, we generate the public parameters:
$$
    \mathbf{A}=
\begin{pmatrix}
6 & 9 & 5 \\\\
8 & 4 & 1 \\\\
5 & 7 & 5 \\\\
\end{pmatrix},
$$
$$
    \mathbf{\check{A}}=
\begin{pmatrix}
6& 9 &5 \\\\
8& 4& 1 \\\\
\end{pmatrix},
$$ and
$$\mathbf{\alpha}= (5,7, 5).$$
In the secure aggregation phase, user 1 obtains the freezing vectors:
$$
\begin{array}{lll}
\mathbf{y}_1^1=\mathbf{\check{A}} \mathbf{x}_1^1=(39,19), \\\\
\mathbf{y}_2^1=\mathbf{\check{A}} \mathbf{x}_2^1=(99,58), \\\\
\mathbf{y}_3^1=\mathbf{\check{A}} \mathbf{x}_3^1=(159,97),
\end{array}
$$
and the key vector:
$$\mathbf{k}^1 =(k_1^1,k_2^1,k_3^1)=(\mathbf{\alpha}\mathbf{x}_1^1,\mathbf{\alpha}\mathbf{x}_2^1,\mathbf{\alpha}\mathbf{x}_3^1)=(34,85,136).$$
User 2 obtains:
$$
\begin{array}{lll}
\mathbf{y}_1^2=\mathbf{\check{A}} \mathbf{x}_1^2=(161,111),\\\\
\mathbf{y}_2^2=\mathbf{\check{A}} \mathbf{x}_2^2=(101,72),\\\\
\mathbf{y}_3^2=\mathbf{\check{A}} \mathbf{x}_3^2=(41,33),
\end{array}
$$
and the key vector is:
$$\mathbf{k}^2 =(k_1^2,k_2^2,k_3^2)=(\mathbf{\alpha}\mathbf{x}_1^2,\mathbf{\alpha}\mathbf{x}_2^2,\mathbf{\alpha}\mathbf{x}_3^2)=(136,85,34).$$
The secure aggregation of the key vectors gives:$$\mathbf{k}^1+\mathbf{k}^2=(170,170,170).$$

For any group (taking the first group as an example), the server cannot deduce $\mathbf{x}^i$ from $\mathbf{y}^i$, but when given $\\sum\mathbf{k}$, the server can solve the following system of equations:
$$
\begin{array}{lll}
A _ {1,1} \sum \mathbf{x}_1+ A _  {1,2} \sum \mathbf{x}_2+A _ {1,3} \sum \mathbf{x}_3=39+161=200 \\\\
A _ {2,1} \sum \mathbf{x}_1+ A _ {2,2}\sum \mathbf{x}_2+A _ {2,3}\sum \mathbf{x}_3=19+111=130 \\\\
A _ {3,1} \sum \mathbf{x}_1+ A _ {3,2}\sum \mathbf{x}_2+A _ {3,3}\sum \mathbf{x}_3=34+136=170
\end{array}
$$
to obtain $\sum \mathbf{x}=(10,10,10)$. Similarly, the aggregated values for other groups can be obtained.

It is evident that Sec. 3.2 and Fig. 4 provide a detailed, general derivation of the above example. Although it may appear complex, each equation is essential for explaining the PVF computation process. We kindly ask the reviewers to read it with patience. If you have any questions, please do not hesitate to raise them.

---

### Author Response · Authors · 2024-11-23
**General Response 2: Significant Progress made during Discussions**

Dear reviewers, PCs and ACs:

We sincerely thank the reviewers for their constructive feedback. After multiple discussions with several reviewers, we summarize the significant progress made regarding our submission as follows:

1. **Privacy** (@Reviewer SAYu, @Reviewer zANE, @Reviewer r5Dg):
   In Appendix D.3, we have refined the proof of Theorem 1, demonstrating that **PVF effectively preserves user privacy**. This point has been **acknowledged by Reviewer r5Dg during the discussion**. We provide the theoretical support indicating that PVF does not leak private elements or their relationships, grounded in the hardness of the Learning With Errors (LWE) decision problem. And we present a detailed security analysis of PVF relying on the Universal Composability (UC) framework. Please refer to page 18 of the latest PDF.

2. **Clarity** (@Reviewer z5KJ, @Reviewer a2uE):
   In General Response 1, we included a concise example to provide a clearer explanation of PVF, which has **aided Reviewer SAYu and Reviewer r5Dg in better understanding PVF**.

If our responses have addressed your concerns, we kindly ask you to consider **raising your scores**. Should any questions remain unresolved or if we have overlooked any of your points, please do not hesitate to raise them.

Thank you very much.

Best regards,

Authors.

---

> ### Comment · Reviewer_r5Dg · 2024-11-23
>
> I would like to differ with the authors in this comment. I have never acknowledged that the proposed method preserves privacy at any point of my discussion and review.

---

> > ### Author Response · Authors · 2024-11-23
> >
> > After our numerous discussions, your latest response led us to believe that we had successfully addressed your concerns. If there has been any misunderstanding, we sincerely apologize. Could you kindly highlight any remaining privacy-related issues with PVF? We will do our best to answer your questions.

---

> > > ### Comment · Reviewer_r5Dg · 2024-11-25
> > >
> > > Dear authors,
> > >
> > > In your response you haven't addressed the main concerns of my last comment: (i.e., https://openreview.net/forum?id=E1Tr7wTlIt&noteId=iog03MH6Wy)
> > >
> > > Therein:
> > > - I explain that satisfying standard security (either universal or sequential composability referenced therein) requires that your protocol don't reveal more information than just the aggregation of the private vectors of the honest parties
> > > - I express my concern in the fact that your protocol reveals overwhelmingly more information than just the sum (i.e. the space of possible solutions of the linear system shrinks exponentially in the number of dimensions of private vectors compared with existing secure protocols).
> > >
> > > Therefore I conclude that your protocol does not satisfy standard security.

---

> > > > ### Author Response · Authors · 2024-11-25
> > > >
> > > > Dear Reviewer r5Dg:
> > > >
> > > > From the comments you provided, we understand that your concerns primarily stem from the possibility that $\mathbf{y}=\check{\mathbf{A}}\mathbf{x}$ in PVF might reveal partial (linear) relationships of $\mathbf{x}$, i.e., information beyond "*the aggregation of the private vectors of the honest parties*". We kindly ask you to review **"Enhanced version: Disrupting Variables Extension (Sec. 3.3)"** in General response 3: Privacy Protection Overview.
> > > >
> > > > In PVF with DVE, $\mathbf{x}^i$ is added with noise $\mathbf{e}$ through Eq. 10 and $\mathbf{y}^i=\check{\mathbf{A}}(\mathbf{x}^i+ \mathbf{e})=\check{\mathbf{A}}\mathbf{x}^i+ \mathbf{e'}$ ($\check{\mathbf{A}}$ is public). Therefore:
> > > > * **The hardness of LWE search problem** ensures that the server cannot obtain any information about $\mathbf{x}^i$ from $\mathbf{y}^i$
> > > > * **The hardness of LWE decision problem** ensures that given a uniformly random vector $\mathbf{w}^i$, $(\check{\mathbf{A}}, \mathbf{y}^i)$ and $(\check{\mathbf{A}}, \mathbf{w}^i)$ are indistinguishable.
> > > >
> > > > That is to say, in PVF with DVE, user privacy **no longer relies on** the hardness of determining a specific solution to an under-determined system of linear equations but instead relies on **the hardness of LWE search and decision problem**. In other words, the phenomenon, where "*the space of possible solutions of the linear system shrinks exponentially in the number of dimensions of private vectors compared with existing secure protocols*", **does not occur** in our scheme.
> > > >
> > > > If there are any aspects of our scheme that lack rigor, we kindly ask you to point them out.

---

> > > > > ### Comment · Reviewer_r5Dg · 2024-11-26
> > > > >
> > > > > Dear authors,
> > > > >
> > > > > Thanks for your reply. Here are some comments about it:
> > > > >
> > > > > The sentence  "similar to many hybrid schemes combining mask and DP (Bonawitz et al., 2017, ..." is not correct as  (Bonawitz et al., 2017)  does not use DP in its protocol.
> > > > >
> > > > > I would like to stress that providing security should not be an "enhancement" of the protocol, but the first basic property. In fact, the goal of Sec 3.3 seems to be to add additional security (a security property that was not yet there as we discussed before) in the case additional entries were compromised (without actually providing a motivation for such scenario). Therefore, the actual goal is currently very confusing.
> > > > >
> > > > > In Eq. (10) you seem yo add Gaussian noise to private values which would somehow simultaneously satisfy DP (for which parameters $\epsilon$ and $\delta$ are not specified) and LWE hardness (which security parameters are also not specified). Note that the variance of the noise would affect both LWE security and DP privacy guarantees. The relation between the type of security and privacy and the variance required to satisfy them is not clear in the paper. From a differential privacy perspective, the variance of the noise require to satisfy acceptable guarantees increases as more linear relations are revealed.
> > > > >
> > > > > In your case, as you reveal almost all linear relations to reconstruct (noisy) private values, the amount of noise that you would need would be similar to local DP, which is huge ($d$ times more than what you would require for secure aggregation that only reveals the sum, where $d$ is the dimension of your private vector). Moreover, the amount of noise required to achieve computational indistinguishabilty is even larger (in fact, much larger). This noise would largely impact your accuracy. In your experiments you seem to set the standard deviation of the noise to $0.040$ pointing to a paper but without explaining where does this come from. As said, from a DP point of view, you would be already required to add a prohibitive amount of noise if the dimension of private vectors is large. None of these precisions are included in your security analysis and no privacy (DP) analysis is done.
> > > > >
> > > > > Finally following your discussion with Reviewer zANE, I would like to stress that I agree with his concerns about malicious security (which, by the way, in the universal composability would require the modelling of an environment that is not in included your proof). Note that the fact that I currently focus in other (important) problems of the paper does not mean that I have not contested this point, which is currently a major issue.

---

> ### Author Response · Authors · 2024-11-27
> **Response part I (Id GR2-6)**
>
> Dear Reviewer r5Dg:
>
> We greatly appreciate your meticulous and insightful feedback, which will be invaluable in refining our work. We hope our response addresses your concerns.
>
> ### **R1. Discussion on Combining DP in PracAgg**
> Please refer to Appendix A in [1]:
>
> *"While secure aggregation alone may suffice for some applications, for other applications stronger guarantees may be needed, as indicated by the failures of ad-hoc anonymization techniques [6, 45, 52], and by the demonstrated capability to extract information about individual training data from fully-trained models (which are essentially aggregates) [26, 50, 51].
> In such cases, secure aggregation composes well with differential privacy [21]. This is particularly advantageous in the local privacy setting [20], which offers provable guarantees for the protection of individual training examples [1, 3] even when the data aggregator is not assumed to be trusted [25, 53]."*
>
> ### **R2. Presentation of Sec. 3.3**
> First, we would like to provide an explanation for the sentence in the original text:
> *"to ensure the privacy of frozen entries even if some secagg entries are compromised."*
> Secagg entries refer to the elements that need to undergo SAP, such as $\mathbf{k}^i$ in General Response 1. Frozen entries refer to $\mathbf{y}^i$. In Sec. 3.2, the security of $\mathbf{x}^i$ relies on the hardness of determining a specific solution to an under-determined system of linear equations, meaning **whether frozen entries can ensure the privacy of $\mathbf{x}^i$ depends on the security of secagg entries**. This is exactly what you have mentioned: the Main PVF may lead to the leakage of correlations between entries. To address this vulnerability, we provide an imporved version in Sec. 3.3.
>
> Regarding the inappropriate use of the term "enhancement", we have revised the phrasing in the latest version of the document.
>
> Furthermore, we have improved the presentation of Sec. 3. In the new PDF, Sec. 3 outlines the entire design process, starting from the **motivation** behind PVF (Sec. 3.1), to the introduction of a **foundational version** (Sec. 3.2), and then to the **improved version** (Sec. 3.3). The foundational version exposes the linear relationship between elements, and thus, in the improved version, we introduce a new method to address this issue. In the improved version, PVF does not disclose any information about $\mathbf{x}^i$ except for $\sum\mathbf{x}^i$.
>
> ### **R3. LWE Parameters**
> It is important to clarify that the noise added in Eq. (10) is **not** based on the principles of Local Differential Privacy (LDP), although it may appear similar. Instead, it is based on Learning With Errors (LWE) (Lemma 3). We have revised the ambiguous parts in the original PDF to eliminate any confusion.
>
> The security parameters of LWE are ($v, q, \sigma$), where $v$ denotes the width of $\check{\mathbf{A}}$, $q$ represents the size of the input space, and $\sigma$ is the standard deviation of $\mathcal{X}$.
> The relationship between the LWE parameters and security can be found in Remark 3 of Appendix D.3 in the latest PDF:
>
> *Regev[2] shows that if the size of $q$ is polynomial in $v$ and $\mathcal{X}$ is a discrete Gaussian distribution on $\mathbb{F}_q$ with standard deviation $\sigma > \frac{2\sqrt{v}}{\sqrt{2\pi}}$, the LWE decision problem is at least as hard as the LWE search problem and solving the LWE search problem can be reduced to solving the Shortest Vector Problem. In DVE, $v=\lambda$, and we use $\mathbb{Z}_p$ as $\mathbb{F}_q$.*
> ### **R4. The standard deviation of $\mathcal{X}$**
> In the original presentation, our focus was on aligning the noise with that added in [3] to assess its impact on model accuracy, specifically by adding noise with the same standard deviation of 0.0409 to the aggregation results (as mentioned in the final paragraph of Sec. 5.1.1 of [3]).
>
> Due to the varying sizes of the input space ($q$) and the matrix width ($v$), adding noise of the same magnitude can impact model accuracy to **different extents**. Intuitively, adding noise of the same magnitude in a 16-bit space will have a much smaller impact on accuracy than in a 32-bit space. Therefore, for consistency, we determine the noise added to the input space according to the noise added to the aggregation result (**in float**) and the security requirements of LWE.
>
> We have included the missing experimental details in the latest PDF (Sec. 5.4):
>
> *In the 32-bit input space, the $\sigma$ is set to 8783 $(>\frac{2 \times 1000}{\sqrt{2\pi}})$, which can meet security requirements in Remark 3 and is equivalent to adding noise with a standard deviation of 0.0409 to the aggregation result (float).*

---

> ### Author Response · Authors · 2024-11-27
> **Response part II (Id GR2-6)**
>
> ### **R5. Modeling malicious participants**
> We have added the modeling of malicious participants in the active adversary model, as detailed in Appendix D.3. Specifically, we outline potential active attacks that a malicious participant might execute during the PVF process and describe how we defend against these attacks using symmetric authenticated encryption and digital signatures.
> ### **Question**
> We are unable to fully comprehend the meaning of the following sentence in your comment:
>
> *"you would be already required to add a prohibitive amount of noise if the dimension of private vectors is large."*
>
> Could you kindly elaborate on the relationship between the required noise magnitude and the dimensionality of the private vectors? Additionally, regarding the relationship between the standard deviation of noise and the hardness of LWE, please refer to **R3** of our response.
>
> If you have any questions, please do not hesitate to raise them.
>
> Authors
>
> ### **Reference**
> [1] Bonawitz, Keith, et al. "Practical secure aggregation for privacy-preserving machine learning." proceedings of the 2017 ACM SIGSAC Conference on Computer and Communications Security. 2017.
>
> [2] Regev, Oded. "On lattices, learning with errors, random linear codes, and cryptography." Journal of the ACM (JACM) 56.6 (2009): 1-40.
>
> [3] Stevens, Timothy, et al. "Efficient differentially private secure aggregation for federated learning via hardness of learning with errors." 31st USENIX Security Symposium (USENIX Security 22). 2022.

---

> > ### Comment · Reviewer_r5Dg · 2024-11-28
> >
> > Dear authors,
> >
> > I have follow-up comments with questions:
> >
> > 1-  Can you please specify which result within the paper of Regev[2] are you using to determine the variance? My comment for this paragraph is that you are still not explaining what security you have. I.e., what is the hardness of solving the Shortest Vector Problem? Every problem also is associated to a security parameter (e.g., "exponential in $\lambda$ where $\lambda=...$). I mean, one can understand the hardness of a problem when used with standard parameters (e.g., large prime groups or fields). However, here, these precisions are not given.
> >
> > 2- There are no details about the parts of (Stevens et al.) [3] that are equivalent or similar to your protocol in such a way that allows you to use similar parameters. Therefore it is still not clear why the final variance of you report would be correct for your protocol. You need to develop why LWE meets the conditions of Regev [2] within your protocol for a given variance and why this variance is correct. In this way it is possible to assess if "$\sigma$ is set to 8783, which can meet security requirements in Remark 3 and is equivalent to adding noise with a standard deviation of 0.0409" is correct. It is not possible to provide an accurate assessment if key elements for the security of the protocol are not provided in the paper.
> >
> > Answer to your question:
> >
> > The noise that would be required for DP for the amount of information that you reveal is very large. DP guarantees are weaker than the kind of indistinguishability provided by LWE security. Therefore, the noise should be larger. This means that if DP noise is already prohibitive, then the noise of your protocol must be even larger.

---

> > > ### Comment · Reviewer_r5Dg · 2024-11-28
> > >
> > > Extra question:
> > >
> > > - What is the exact requirement of "the size of $q$ is polynomial in $v$"? $v$ here is a variable that changes with the learning task. However, $q$ seems to be fixed to 32 in your experiments. I get the impression that the concrete security of your protocol may depend on $q$.

---

> > > > ### Author Response · Authors · 2024-11-29
> > > > **Response (Id GR2-10)**
> > > >
> > > > Firstly, q (i.e., p in PVF) is **certainly not 32**. As is well known, $\mathbb{Z}_p$ refers to the set $\\{ 0, 1, \ldots, p-1 \\}$ for a large prime p. In this paper, the input space is 32-bit, meaning p is a **fixed 32-bit large prime**, specifically 4294967291.
> > > >
> > > > Secondly, in this paper, $v = \lambda$, and $\lambda$ can be fixed to a value in $[100, 1000]$.
> > > >
> > > > Finally, for the security requirements of PVF, please refer to Remark 3.

---

> > > > > ### Comment · Reviewer_r5Dg · 2024-11-29
> > > > >
> > > > > Dear authors,
> > > > >
> > > > > I apologize for the confusion of $q$. By $32$ I meant 32 bits.
> > > > >
> > > > > Thanks for your answers. However your response on the use of the results of Regev's paper does not convince me for the following reason:
> > > > >
> > > > > The main theorem (Thm 3.1) of Regev's paper is about the hardness of solving $LWE_{p,\Psi}$ given a polynomial size sets of samples of the distribution  $(A, Ax +e)$ where
> > > > >    - $A$ is chosen uniformly at random
> > > > >    - $e$ has a Gaussian distribution $\Psi$
> > > > >
> > > > > Note that in the paper, the distribution of $e$ is independent of the matrix $A$. Therefore, in your paper, if $\check{A}$ is also a random variable you cannot use $e' = \check{A}e$ where $e$ is Gaussian because $e'$ is not Gaussian (i.e., only the conditional distribution of $e'$ given $\check{A}$ follows a Gaussian distribution, but not $e'$ by itself). Therefore, it is my impression that you cannot apply this theorem.
> > > > >
> > > > > I stress that these aspects are very important for the security of the protocol and none of them have been explained in the manuscript.

---

> > > > > > ### Author Response · Authors · 2024-11-29
> > > > > > **Response (Id GR2-12)**
> > > > > >
> > > > > > Dear Reviewer r5Dg:
> > > > > >
> > > > > > Please refer to **line 270 (in Sec. 3.3)**, **Fig. 13**, **General Response 3**, or [this comment](https://openreview.net/forum?id=E1Tr7wTlIt&noteId=d9RgcLFX4B), where we have repeatedly emphasized that $\mathbf{A}$, including $\check{\mathbf{A}}$, is **public** (as is $\mathbf{A}$ in LWE) and remains **unchanged** throughout the entire training task. $\check{\mathbf{A}}$ is a **constant**, and therefore, $\mathbf{e}'$ remains a random noise vector following the Gaussian distribution.
> > > > > >
> > > > > > If you have any questions, please do not hesitate to raise them.

---

> ### Author Response · Authors · 2024-11-29
> **Response (Id GR2-8)**
>
> Dear Reviewer r5Dg:
>
> Based on the questions you raised, we speculate that you may not have thoroughly reviewed the latest version of our PDF. In the following response, we will include relevant excerpts from the PDF where necessary.
>
> ### **R1. The theoretical foundation of Remark 3 in our paper**
> Remark 3 in our paper summarizes the content of "*THEOREM 3.1 (MAIN THEOREM)*" on page 20 of [1]. In simple terms, Regev demonstrated that when $\alpha q > 2\sqrt{v}$, or equivalently $\sigma > \frac{2\sqrt{v}}{\sqrt{2\pi}}$, where $\sigma = \frac{\alpha q}{\sqrt{2\pi}}$ (see II.B in [2]), solving the LWE problem would be equivalent to solving the Shortest Vector Problem, a well-known NP-hard problem. In our experiment (Fig. 6), $\sigma$ is set to 8783 $(>\frac{2 \times \sqrt{1000}}{\sqrt{2\pi}})$, which satisfies the security requirements.
> ### **R2. The standard deviation of $\mathcal{X}$**
> Please refer to the description in R1. The standard deviation of the added noise must satisfy $(>\frac{2 \times \sqrt{\lambda}}{\sqrt{2\pi}})$. In our experiments, $\lambda$ ranges from [100, 1000], so the **maximum** required minimum standard deviation is $\frac{2 \times \sqrt{1000}}{\sqrt{2\pi}} = 25$. We set it to 8783 merely to demonstrate that even when noise far exceeding the security requirements is added, the impact on the model's accuracy remains negligible.
>
> ### **About the Question**
> *"you would be already required to add a prohibitive amount of noise if the dimension of private vectors is large."*
>
> We still do not understand this comment based on the response you provided. We have emphasized that the security foundation of this paper is **based on LWE, not DP**. Although both DP and LWE involve Gaussian noise, their principles for achieving privacy protection are **entirely different**. The LWE problem and the conditions under which it is difficult have been **clearly explained** in our paper. We kindly ask the reviewer to consult the definitions of DP and LWE and compare them to understand the differences between the two. Furthermore, based on your intuitive and simple explanation, we do not understand why the noise intensity in our proposed scheme (or in LWE problem) would be related to the vector dimension. We would appreciate it if the reviewer could provide a **formal** explanation or an **example** to clarify this, or perhaps refer to **relevant literature and theoretical foundations** to support this claim.
>
> If you have any questions, please do not hesitate to raise them.
>
> ### Reference
> [1] Regev, Oded. "On lattices, learning with errors, random linear codes, and cryptography." Journal of the ACM (JACM) 56.6 (2009): 1-40.
>
> [2] Marcolla, Chiara, et al. "Survey on fully homomorphic encryption, theory, and applications." Proceedings of the IEEE 110.10 (2022): 1572-1609.

---

> ### Comment · Reviewer_r5Dg · 2024-11-29
>
> I have already read that $\check{A}$ is public in the paper and in your previous comment. However, it does not mean that your use of Theorem 3.1 is correct.
>
> Let me be more specific:
>
> The LWE problem stated in Regev's paper defines observations as  pairs $(\mathbf{a}, \mathbf{a}^\top \mathbf{x} + e)$ where $\mathbf{x}$ is your secret vector of dimension $\lambda$,  $\mathbf{a}$ is a random vector sampled uniformly at random from $\mathbb{Z}_p^{\lambda}$ and $e$ is a scalar sampled from the gaussian distribution $\Psi$. For all observations $e$'s are i.i.d samples of a given $\Psi$.
>
> In your paper you have a matrix $\check{A} \in \mathbb{Z}\_p^{(\lambda-1) \times \lambda }$. The pair $(\check{A}, \check{A}\mathbf{x} + \mathbf{e}')$ cannot be used as a single observation, because observations are vector-vector products and not matrix-vector products. This in itself is not a problem as you can define the set of $\lambda-1$ observations of the form $(\check{A}\_{i,:}, \check{A}\_{i,:}\mathbf{x} + \mathbf{e}'\_i )$, where for all $i \in \\{1,\dots, \lambda-1\\}$, $\check{A}\_{i,:}$ is the $i$th row of $\check{A}$ and $\mathbf{e}'\_i = \check{A}_{i,:} \mathbf{e}$ for a vector $\mathbf{e}$ that follows a multivariate Gaussian distribution.
>
> Lets examine the distribution of your observations. Each vector $\check{A}\_{i,:}$ is a random vector as $\check{A}$ is a random matrix in the adequate domain. However, your distribution of pairs cannot be stated as the distributions of the paper for the following reasons:
> - each sample $\mathbf{e}'\_i$  has a different Gaussian distribution dependent on $\check{A}_{i,:}$, while in Regev's paper they should all be samples from the same distribution
> - all samples  $\mathbf{e}'\_i$ are correlated with each other, as they all depend on $\mathbf{e}$. However, in Regev's paper all samples of $\mathbf{e}'\_i$ must be independent.
>
> Therefore, your protocol does not appear to meet the preconditions to apply Theorem 3.1 of Regev's paper.

---

> > ### Author Response · Authors · 2024-12-01
> > **Response (Id GR2-14)**
> >
> > Dear Reviewer r5Dg:
> >
> > Thank you for your more detailed explanation. We hope the following response will address your concerns.
> >
> > ### **Notation**
> > We consider $\langle\mathbf{A}, \mathbf{e}\rangle$ to be an LWE instance, and $\langle \mathbf{a _ i}, \mathbf{a _ i x}+e _ i \rangle$ represents a sample from the instance $\langle\mathbf{A}, \mathbf{e}\rangle$.
> > ### **R1: Response to the First Concern**
> > Each $e$ follows a different Gaussian distribution **does not** affect the hardness of the LWE problem. For instance, consider $<\mathbf{a}, \mathbf{ax} + e>$ and $<\mathbf{a}, \mathbf{ax} + e'>$. These can be viewed as belonging to **different** LWE instances, which does not compromise security. Intuitively, providing $\lambda$ samples from an instance $<\mathbf{A}, \mathbf{e}>$ does not affect its security, and giving just a single sample will similarly have no impact on security.
> >
> > ### **R2: Response to the Second Concern**
> >
> > This issue can be easily resolved by making **a slight adjustment to the distribution of $\mathbf{e}$**. Before we begin, Let us present a property of the multivariate Gaussian distribution:
> >
> > *If a random vector $\mathbf{X}$ follows a multivariate Gaussian distribution, the zero elements of its covariance matrix indicate that the corresponding components are independent of each other.*
> >
> >
> > Let's consider the general scenario: given $l (> \lambda)$ LWE samples, i.e., $\langle \mathbf{a} _ i, \mathbf{a} _ i \mathbf{x} + e _ i' \rangle$, for $i \in [1, l]$. Based on fundamental knowledge of linear systems, it is clear that, aside from the $\lambda$ LWE samples with linearly independent $\mathbf{a} _ i$, the remaining samples are redundant. Therefore, we focus our analysis on the $\lambda$ LWE samples with linearly independent $\mathbf{a} _ i$, i.e., $\mathbf{A} \in \mathcal{Z} _ p^{\lambda \times \lambda}$.
> >
> > Given $\mathbf{A}$ (**the public parameter**, i.e., $\lambda$ linearly independent $\mathbf{a} _ i$), we prove that when $\mathbf{e} \sim \mathcal{N}(0, \Sigma _ e=\mathbf{A}^{-1}\mathbf{D}\mathbf{A}^{-T})$, where $\Sigma _ e$ is the covariance matrix of $\mathbf{e}$ and $\mathbf{D}$ is a diagonal matrix, the elements of $\mathbf{e}'=\mathbf{A} \mathbf{e}$ are independent:
> >
> > Proof.
> > $\mathbf{e}'$ follows a multivariate normal distribution, with its mean and covariance matrix given by:
> >
> > $$
> > \mathbb{E}[\mathbf{e}'] = \mathbb{E}[\mathbf{A}\mathbf{e}] = \mathbf{A}\mathbb{E}[\mathbf{e}] = \mathbf{A} \cdot 0 = 0
> > $$
> > $$
> > \text{Cov}(\mathbf{e}') = \mathbf{A}\text{Cov}(\mathbf{e}) \mathbf{A}^T = \mathbf{A} \Sigma _ e \mathbf{A}^T= \mathbf{A} \mathbf{A}^{-1}\mathbf{D}\mathbf{A}^{-T} \mathbf{A}^T=\mathbf{D}
> > $$
> >
> > Therefore, each component of $\mathbf{e}'$ **independently** follows a Gaussian distribution.
> >
> > When **selecting any** $\lambda - 1$ or fewer LWE samples from the aforementioned $\lambda$ samples, the added noise is also **independent** of each other and the amount of information obtainable is **even less** than that from all $\lambda$ samples, thereby ensuring that privacy is not compromised.
> >
> > After the above adjustments, the standard deviation of the noise added to $x _ i$ is $\sigma _ i = (\mathbf{A}^{-1}\mathbf{D}\mathbf{A}^{-T}) _ {ii} = \sigma _ i' \sum _ {k=1}^\lambda (\mathbf{A}^{-1}) _ {ik}^2$. Since $p$ is a prime number, i.e., $\gcd(\sum _ {k=1}^\lambda (\mathbf{A}^{-1}) _ {ik}^2, p) = 1$, the map $\sigma _ i' \mapsto \sigma _ i' \sum _ {k=1}^\lambda (\mathbf{A}^{-1}) _ {ik}^2 \mod p$ is a bijective mapping from $\mathbb{Z} _ p$ to $\mathbb{Z} _ p$. $\sigma _ i'$ must satisfy $\sigma _ i' > 2\sqrt{\lambda}$, which means that the range of invalid values for $\sigma _ i'$ is **extremely small**. Therefore, $\sigma _ i' \sum _ {k=1}^\lambda (\mathbf{A}^{-1}) _ {ik}^2$ can basically take all values ​​of $\mathbb{Z} _ p$.
> > It is evident that we can always choose a $\sigma _ i'$ such that
> > $$\sigma _ i' \sum _ {k=1}^\lambda (\mathbf{A}^{-1}) _ {ik}^2 \mod p < \epsilon,$$
> > where $\epsilon$ is a small number representing the maximum noise that can be added to $x _ i$.
> >
> > For example, in Fig. 6, the range of invalid values for $\sigma _ i'$ is $[0, \frac{2\sqrt{1000}}{\sqrt{2\pi}} = 25]$, meaning that there are **only 26** specific values in $\mathbb{Z} _ p$ that $\sigma  _ i' \sum _ {k=1}^\lambda (\mathbf{A}^{-1}) _ {ik}^2$ cannot take. Since $\epsilon = 8783$ still ensures that the added noise is negligible and there are only 26 values that cannot be taken, $\sigma _ i' \sum _ {k=1}^\lambda (\mathbf{A}^{-1}) _ {ik}^2 \mod p$ can always take values within the range $[0, 8783]$. According to **R1**, the fact that the diagonal elements of $\mathbf{D}$ are different does not compromise security. Thus, we can always construct a $\mathbf{D}$ that not only ensures the noise added to $\mathbf{x}$ is negligible but also guarantees that the components of $\mathbf{e}'$ in the LWE instance are independent of each other and $\sigma _ i' > 2\sqrt{\lambda}$.

---

> > > ### Comment · Reviewer_r5Dg · 2024-12-03
> > >
> > > Dear authors,
> > >
> > > Thank you for your detailed reply and efforts in modifying the manuscript.
> > >
> > > I am however still not convinced that the distribution of observations in your protocol is the same as the distribution required in Theorem 3.1 of (Regev. et al.).
> > >
> > > 1- For $\mathbf{e}'$ to have a Gaussian distribution, you have to fix $A$ in advance. However, in  (Regev. et al.) observations of the form $(\mathbf{a}, \mathbf{a}^\top \mathbf{x} + e_i)$ (as described in my previous e-mail) require that $\mathbf{a}$ follows a random distribution, which is not the case in your protocol as now $A$ is fixed.
> > >
> > > 2- Your last modification of the protocol seems to make $\mathbf{e}'$ independent, but now $\mathbf{e}$ is correlated.This effect needs to be proven secure in the overall revealed information: In addition to  $\check{A}\mathbf{x}$ that each user reveals, you will reveal the aggregation of all private values with a correlated noise term. This might not be innocuous. Please try to provide a step-by-step proof.
> > >
> > > Given that we are at the end of the discussion period, I feel that it is hard for reviewers to assess the changes that you already added and I still think that substantial changes on other problems that were mentioned in the discussion need to be made to the protocol, including a proper security proof. A friendly guide to do that is provided in [1].
> > >
> > > I will keep my score because the current manuscript requires a large amount of work for its acceptance, but I encourage authors to keep investigating the security of this idea.
> > >
> > > [1] Lindell, Yehuda. "How to simulate it–a tutorial on the simulation proof technique." Tutorials on the Foundations of Cryptography: Dedicated to Oded Goldreich (2017): 277-346.

---

> > > > ### Author Response · Authors · 2024-12-03
> > > > **Response (Id GR2-16)**
> > > >
> > > > Dear Reviewer r5Dg:
> > > >
> > > > ### **R1: Public $\mathbf{A}$ Does Not Compromise Security**
> > > > In our scheme, $\mathbf{A}$ is public, **does not** require manual construction, and is randomly generated. Familiarity with public key cryptosystems might aid in understanding this concept. You may refer to *Public Key Cryptosystem* of [1] on page 35, where $\mathbf{a}_i$ is **also** used as the *Public Key*. Similarly, in their scheme, $\mathbf{a}_i$ is also "**fixed in advance**."
> > > >
> > > > ### **R2: Privacy of Aggregation Results**
> > > > Firstly, we have demonstrated that $\mathbf{\check{A}x}$ does not reveal any private information.
> > > >
> > > > Secondly, even without adding any noise, as shown in [2], $\sum \mathbf{x}^i$ **does not** compromise the privacy of any individual $\mathbf{x}^i$, which is a foundational principle and consensus in **secure multi-party computation**. Consequently, adding noise (which itself is an aggregated value) does not affect the privacy of the values participating in the aggregation.
> > > >
> > > > In our discussions, we have thoroughly demonstrated the security of our scheme and addressed all the concerns you previously raised. Could you please **specify** why you believe "the distribution of observations in your protocol is the same as the distribution required in Theorem 3.1 of [1]"?
> > > >
> > > > Additionally, the current version of the PDF already includes an almost complete security proof, and we only need to incorporate the content of Response Id GR2-14 into the PDF. This discussion has taken place under the General Response, with other reviewers observing the entire process. **We do not believe that making minor adjustments at the end of the discussion period should serve as grounds for rejecting our work**. If we have adequately resolved your concerns, we earnestly hope that you will consider raising your score.
> > > >
> > > > ### Reference
> > > > [1] Regev, Oded. "On lattices, learning with errors, random linear codes, and cryptography." Journal of the ACM (JACM) 56.6 (2009): 1-40.
> > > >
> > > > [2] Bonawitz, Keith, et al. "Practical secure aggregation for privacy-preserving machine learning." proceedings of the 2017 ACM SIGSAC Conference on Computer and Communications Security. 2017.

---

> ### Comment · Reviewer_SAYu · 2024-12-03
>
> Dear authors and reviewer r5Dg,
>
> Thanks for your detailed discussion and sorry for interrupting the conversation.
>
> I also agree with the reviewer r5Dg, i.e., I am also not convinced that the distribution of observations in your protocol is the same as the distribution required in Theorem 3.1 of (Regev. et al.).
>
> I think the misunderstanding stems from the author's statement "$\sum{\mathbf{x}}^{i}$ does not compromise the privacy of any individual ${\mathbf{x}}^{i}$, which is a fundational principle and consensus in secure multi-party computation", which is not true.
> The sum of the local models definitely includes some information about the local model, and the amount of leaked information in secure aggregation protocol is $O(1/N)$ where $N$ is the number of models to be aggregated [1].
>
> [1] Elkordy, A., et al. "How Much Privacy Does Federated Learning with Secure Aggregation Guarantee?." PETS. 2023.

---

> > ### Author Response · Authors · 2024-12-03
> > **Response (Id GR2-18)**
> >
> > Dear Reviewer SAYu:
> >
> > Thanks for your reply.
> >
> > The primary concern does not stem from the statement "$\sum \mathbf{x}^i$ does not compromise the privacy of any individual $\mathbf{x}^i$, which is a foundational principle and consensus in secure multi-party computation." This is a privacy guarantee for secure aggregation, **not** the theoretical foundation of our scheme. We have consistently focused on demonstrating that PVF **does not compromise the security** of secure aggregation.
> >
> > Investigating vulnerabilities and defenses within secure aggregation protocols is beyond the scope of our work. As we pointed out in our document (line 105):
> >
> > *Note that the security of FL remains an open issue. SAPs, though unable to fully guarantee FL security at present, remain a promising direction worth exploring. The main objective of our work is to **reduce the masking-related overhead of secure aggregation, thereby making it more applicable in practice**.*

---

### Author Response · Authors · 2024-11-23
**General response 3: Privacy Protection Overview**

Dear reviewers, PCs and ACs:

The primary concerns raised by @Reviewer SAYu, @Reviewer zANE, and @Reviewer r5Dg revolve around the potential leakage of partial information about $\mathbf{x}^i$ through $\mathbf{y}^i$, leading to privacy compromise. Below, we briefly outline the privacy protection of our approach to substantiate the claim that "*PVF effectively preserves user privacy*" in General Response 2:
### **1. Basic version: privacy in the Main PVF method (Sec. 3.2)**
The privacy of $\mathbf{x}^i$ in the Main PVF is primarily safeguarded by **the hardness of determining a specific solution to an under-determined system of linear equations**. For instance, given the private vector $\mathbf{x} = (x_1, x_2, x_3) = (7, 0, 0)$ and the matrix $\mathbf{A}$ in General Response 1:
$$
    \mathbf{A}=
\begin{pmatrix}
6 & 9 & 5 \\\\
8 & 4 & 1 \\\\
5 & 7 & 5 \\\\
\end{pmatrix},
$$
From the server's perspective, it only has access to $\mathbf{y}$, which is expressed as:
$$
\begin{array}{lll}
6x_1+9x_2+5x_3=42, \\\\
8x_1+4x_2+x_3=56,
\end{array}
$$
The server cannot distinguish whether $\mathbf{x}$ is $(7, 0, 0)$ or $(18, -34, 48)$ or any other possible solution. While the server cannot obtain any individual element of $\mathbf{x}$, it still obtains certain linear relationships involving the private elements, as pointed out by the reviewers. For this reason, we introduced Disrupting Variables Extension (DVE) in Sec. 3.3 to provide enhanced privacy.
### **2. Enhanced version: Disrupting Variables Extension (Sec. 3.3)**
DVE ensures that **the server cannot obtain any information about $\mathbf{x}^i$ from $\mathbf{y}^i$**, relying on **the hardness of the Learning With Errors (LWE) decision problem**. As detailed in Appendix D.3 (Lemma 3: The hardness of the Learning With Errors decision problem):

*Given a finite field $\mathbb{F}_p$ and a discrete probability distribution $\mathcal{X}$ over $\mathbb{F}_p$. Let $\mathbf{s} \in \mathbb{F}_p^n$ be a secret vector, $\mathbf{A} \in \mathbb{F}_p^{m \times n}$ be a matrix that is chosen uniformly at random and $\mathbf{e} \in \mathbb{F}_p^m$ be the error vector that is sampled from $\mathcal{X}$. The Learning With Errors (LWE) (search) problem is to find $\mathbf{s}$, given the pair $(\mathbf{A}, \mathbf{b})$, where $\mathbf{b} = \mathbf{A} \mathbf{s} + \mathbf{e}$. And the LWE decision problem is to distinguish between two uniformly randomly generated pairs. When the size of $p$ is polynomial in $n$, the LWE decision problem is at least as hard as the LWE search problem.*

$\mathbf{x}^i$ is added with noise $\mathbf{e}$ through Eq. 10 and $\mathbf{y}=\check{\mathbf{A}}(\mathbf{x}+ \mathbf{e})=\check{\mathbf{A}}\mathbf{x}+ \mathbf{e'}$ ($\check{\mathbf{A}}$ is public). Therefore, given a uniformly random vector $\mathbf{w}^i$, Lemma 3 ensures that $(\check{\mathbf{A}}, \mathbf{y}^i)$ and $(\check{\mathbf{A}}, \mathbf{w}^i)$ are indistinguishable, which guarantees $\mathcal{S}$ does not obtain private information from honest users through frozen vectors.

### **3. Overall security analysis**
Finally, we performed a security analysis of the secure aggregation protocol integrated with PVF under the Universal Composability (UC) framework, i.e., the proof of Theorem 1.

We sincerely invite you to review our responses and hope they resonate with your insights. Your dedication and expertise in evaluating our work have been invaluable in enhancing its quality.

Best regards,

Authors.

---

### Author Response · Authors · 2024-12-03
**General response 4: Looking Forward to Your Reply**

Dear reviewers, PCs and ACs:

We are immensely grateful to the reviewers for their valuable feedback, as well as to the PCs and ACs for their coordination efforts. The discussions over the past three weeks have yielded numerous insightful suggestions, which have been instrumental in improving the quality of our work.
Following several discussions with multiple reviewers, we have now responded to all the questions and concerns raised. With **fewer than 12 hours** remaining until the conclusion of the discussion period, we sincerely invite you to participate in the dialogue. If you have any further questions or new insights, your input would be greatly valued. **If we have already addressed your concerns to your satisfaction, we kindly request that you consider raising our score.**

**In General Response 3**, we demonstrated that PVF effectively protect user privacy. Furthermore, through extensive discussions with Reviewer r5Dg (to whom we express our sincere respect and gratitude), we have thoroughly examined and provided detailed explanations regarding privacy protection (please refer to **Responses GR2-6 to GR2-14**). Since PVF significantly reduces the overhead of secure aggregation, we are confident that our work will contribute greatly to both community research and practical engineering applications.

Best regards,

Authors.

---

### Meta-Review · Area_Chair_g6rG · 2024-12-05

**Metareview:**

The paper proposes $\lambda$-SecAgg, a secure aggregation protocol for federated learning (FL), to reduce computational and communication overhead using Partial Vector Freezing (PVF). However, most reviewers raised their concerns that PVF significantly reduces this privacy. Despite extensive discussion on these issues, reviewers have maintained their reviews and scores. Given these issues, I recommend rejection.

**Additional Comments On Reviewer Discussion:**

Most reviewers noted that this work lacks a standard notion of security. The authors provided a more detailed explanation but failed to address reviewers' concerns. I agree with Reviewer r5Dg that this paper still requires a large amount of work for its acceptance.

---

### Decision · Program_Chairs · 2025-01-22

Reject